# Embedding Fourier for Ultra-High-Definition Low-Light Image Enhancement

**Chongyi Li**[1]**, Chun-Le Guo**[2]**, Man Zhou**[1]**, Zhexin Liang**[1]**,**
**Shangchen Zhou**[1]**, Ruicheng Feng**[1]**, and Chen Change Loy**[1]
[1]S-Lab, Nanyang Technological University  [2]Nankai University

## Abstract

Ultra-High-Definition (UHD) photo has gradually become the standard configuration in advanced imaging devices. The new standard unveils many issues in existing approaches for low-light image enhancement (LLIE), especially in dealing with the intricate issue of joint luminance enhancement and noise removal while remaining efficient. Unlike existing methods that address the problem in the spatial domain, we propose a new solution, **UHDFour**, that embeds Fourier transform into a cascaded network. Our approach is motivated by a few unique characteristics in the Fourier domain: 1) most luminance information concentrates on amplitudes while noise is closely related to phases, and 2) a high-resolution image and its low-resolution version share similar amplitude patterns. Through embedding Fourier into our network, the amplitude and phase of a low-light image are separately processed to avoid amplifying noise when enhancing luminance. Besides, UHDFour is scalable to UHD images by implementing amplitude and phase enhancement under the low-resolution regime and then adjusting the high-resolution scale with few computations. We also contribute the first real UHD LLIE dataset, **UHD-LL**, that contains 2,150 low-noise/normal-clear 4K image pairs with diverse darkness and noise levels captured in different scenarios. With this dataset, we systematically analyze the performance of existing LLIE methods for processing UHD images and demonstrate the advantage of our solution. We believe our new framework, coupled with the dataset, would push the frontier of LLIE towards UHD. The code and dataset are available at https://li-chongyi.github.io/UHDFour/.

## 1 Introduction

With the advent of advanced imaging sensors and displays, Ultra-High-Definition (UHD) imaging has witnessed rapid development in recent years. While UHD imaging offers broad applications and makes a significant difference in picture quality, the extra pixels also challenge the efficiency of existing image processing algorithms.

In this study, we focus on one of the most challenging tasks in image restoration, namely low-light image enhancement (LLIE), where one needs to jointly enhance the luminance and remove inherent noises caused by sensors and dim environments. Further to these challenges, we lift the difficulty by demanding efficient processing in the UHD regime.

Despite the remarkable progress in low-light image enhancement (LLIE) (Li et al., 2021a), existing methods (Zhao et al., 2021; Wu et al., 2022; Xu et al., 2022), as shown in Figure 1, show apparent drawbacks when they are used to process real-world UHD low-light images. This is because (1) most methods (Guo et al., 2020; Liu et al., 2021b; Ma et al., 2022) only focus on luminance enhancement and fail in removing noise; (2) some approaches (Wu et al., 2022; Xu et al., 2022) simultaneously enhance luminance and remove noise in the spatial domain, resulting in the suboptimal enhancement; and (3) existing methods (Wei et al., 2018; Zhao et al., 2021; Wu et al., 2022; Xu et al., 2022) are mainly trained on low-resolution (LR) data, leading to the incompatibility with high-resolution (HR) inputs; and (4) some studies (Xu et al., 2022; Zamir et al., 2022) adopt heavy structures, thus being inefficient for processing UHD images. More discussion on related work is provided in the Appendix.

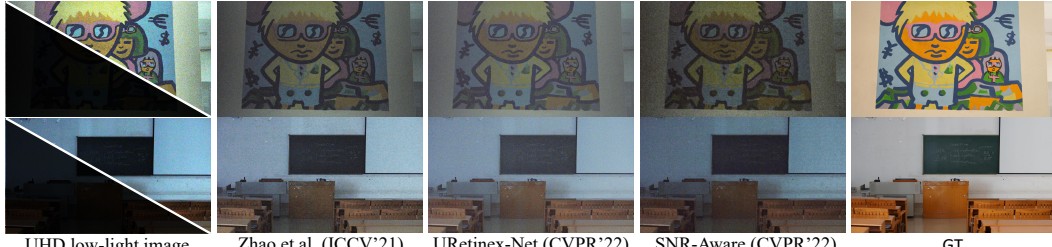

UHD low-light image     Zhao et al. (ICCV'21)     URetinex-Net (CVPR'22)     SNR-Aware (CVPR'22)     GT

**Figure 1:** Visual results of state of the arts (Zhao *et al.* (Zhao et al., 2021), URetinex-Net (Wu et al., 2022), and SNR-Aware (Xu et al., 2022)) pre-trained on an existing low-light image dataset for processing the real-world UHD low-light images. We amplify the brightness of the input UHD low-light images 10 times (top right corner of the first column) to show details and noise. These officially released models were trained using existing paired LR images with mild noise (i.e., the LOL dataset (Wei et al., 2018)). Existing models cannot cope with challenging UHD low-light images well.

To overcome the challenges aforementioned, we present a new idea for performing LLIE in the Fourier Domain. Our approach differs significantly from existing solutions that process images in the spatial domain. In particular, our method, named as **UHDFour**, is motivated by our observation of two interesting phenomena in the Fourier domain of low-light noisy images: i) luminance and noise can be decomposed to a certain extent in the Fourier domain. Specifically, luminance would manifest as amplitude while noise is closely related to phase, and ii) the amplitude patterns of images of different resolutions are similar. These observations inspire the design of our network, which handles luminance and noise separately in the Fourier domain. This design is advantageous as it avoids amplifying noise when enhancing luminance, a common issue encountered in existing spatial domain-based methods. In addition, the fact that amplitude patterns of images of different resolutions are similar motivates us to save computation by first processing in the low-resolution regime and performing essential adjustments only in the high-resolution scale.

We also contribute the first benchmark for UHD LLIE. The dataset, named **UHD-LL**, contains 2,150 low-noise/normal-clear 4K UHD image pairs with diverse darkness and noise levels captured in different scenarios. Unlike existing datasets (Wei et al., 2018; Lv et al., 2021; Bychkovsky et al., 2011) that either synthesize or retouch low-light images to obtain the paired input and target sets, we capture real image pairs. During data acquisition, special care is implemented to minimize geometric and photometric misalignment due to camera shake and dynamic environment. With the new UHD-LL dataset, we design a series of quantitative and quantitative benchmarks to analyze the performance of existing LLIE methods and demonstrate the effectiveness of our method.

Our contributions are summarized as follows: **(1)** We propose a new solution for UHD LLIE that is inspired by unique characteristics observed in the Fourier domain. In comparison to existing LLIE methods, the proposed framework shows exceptional effectiveness and efficiency in addressing the joint task of luminance enhancement and noise removal in the UHD regime. **(2)** We contribute the first UHD LLIE dataset, which contains 2,150 pairs of 4K UHD low-noise/normal-clear data, covering diverse noise and darkness levels and scenes. **(3)** We conduct a systematical analysis of existing LLIE methods on UHD data.

## 2    OUR APPROACH

In this section, we first discuss our observations in analyzing low-light images in the Fourier domain, and then present the proposed solution.

### 2.1    OBSERVATIONS IN FOURIER DOMAIN

Here we provide more details to supplement the observations we highlighted in Sec. 1. We analyze real UHD low-light images in the Fourier domain and provide a concise illustration in Figure 2. Specifically, (a) Swapping the amplitude of a low-light and noisy (low-noise) image with that of its corresponding normal-light and clear (normal-clear) image produces a normal-light and noisy (normal-noise) image and a low-light and clear (low-clear) image. We show more examples in the Appendix. The result suggests that the luminance and noise can be decomposed to a certain extent in the Fourier domain. In particular, most luminance information is expressed as amplitudes, and

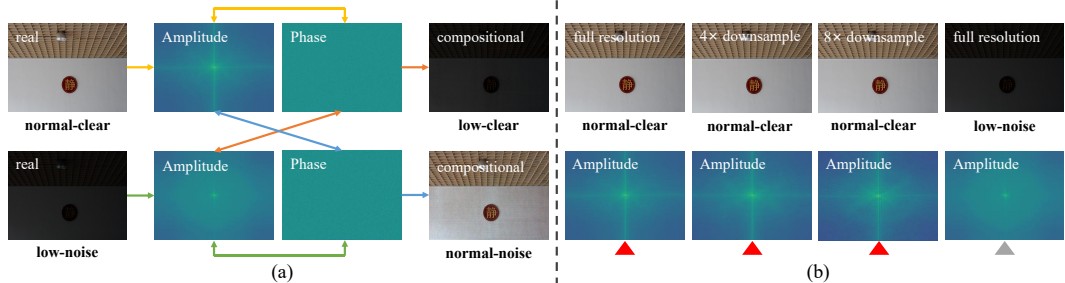

**Figure 2:** Motivations. We observed that **(a)** luminance and noise can be 'decomposed' to a certain extent in the Fourier domain and **(b)** HR image and its LR versions share similar amplitude patterns. The amplitude and phase are produced by Fast Fourier Transform (FFT) and the compositional images are obtained by Inverse FFT (IFFT). For visualization, we show the amplitude and phase in imagery format with common transformations. Lines of the same color indicate a set of FFT/IFFT transforms. The red triangles mark the similar pattern (obviously different from the gray one). Zoom in for the details and noise. We show more examples and analysis in the Appendix.

noises are revealed in phases. This inspires us to process luminance and noise separately in the Fourier domain. (b) The amplitude patterns of an HR normal-clear image and its LR versions are similar and are different from the corresponding HR low-noise counterpart. Such a characteristic offers us the possibility to first enhance the amplitude of an LR scale with more computations and then only make minor adjustments in the HR scale. In this way, most computations can be conducted in the LR space, reducing the computational complexity.

## 2.2 THE UHDFOUR NETWORK

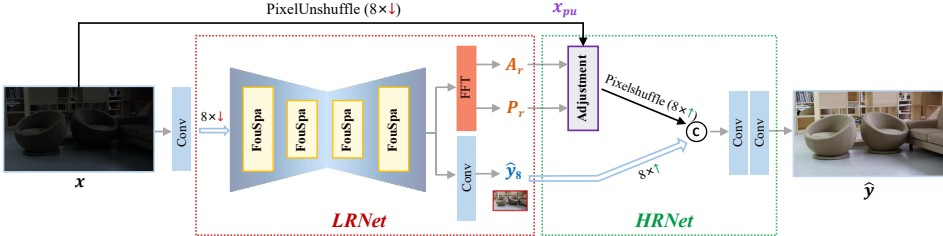

**Figure 3:** Overview of UHDFour. Our approach consists of an LRNet and an HRDNet. The LRNet is an encoder-decoder network that produces $8\times$ downsampled result $\hat{y}_8$ and the refined amplitude $A_r$ and phase $P_r$ features. We omit the skip connections for brevity. The HRNet contains an Adjustment Block and the upsampling operation, producing the final result $\hat{y}$. Most computation is conducted in the LRNet.

**Overview.** UHDFour aims to map an UHD low-noise input image $x \in \mathbb{R}^{H \times W \times C}$ to its corresponding normal-clear version $y \in \mathbb{R}^{H \times W \times C}$, where $H$, $W$, and $C$ represent height, width, and channel, respectively. Figure 3 shows the overview of UHDFour. It consists of an LRNet and an HRNet.

Motivated by the observation in Sec. 2.1, LRNet takes the most computation of the whole network. Its input is first embedded into the feature domain by a Conv layer. To reduce computational complexity, we downsample the features to 1/8 of the original resolution by bilinear interpolation. Then, the LR features go through an encoder-decoder network, which contains four FouSpa Blocks with two $2\times$downsample and two $2\times$upsample operations, obtaining outputs features. The outputs features are respectively fed to FFT to obtain the refined amplitude $A_r$ and phase $P_r$ features and a Conv layer to estimate the LR normal-clear image $\hat{y}_8 \in \mathbb{R}^{H/8 \times W/8 \times C}$.

The outputs of LRNet coupled with the input are fed to the HRNet. Specifically, the input $x$ is first reshaped to $x_{pu} \in \mathbb{R}^{H \times W \times C \times 64}$ via PixelUnshuffle ($8\times \downarrow$) to preserve original information, and then fed to an Adjustment Block. With the refined amplitude $A_r$ and phase $P_r$ features, the Adjustment Block produces adjusted features that are reshaped to the original height and width of input $x$ via Pixelshuffle ($8\times \uparrow$). Finally, we resize the estimated LR normal-clear image $\hat{y}_8$ to the original size of input $x$ via bilinear interpolation and combine it with the upsampled features to estimate the final HR normal-clear image $\hat{y}$. We detail the key components as follows.

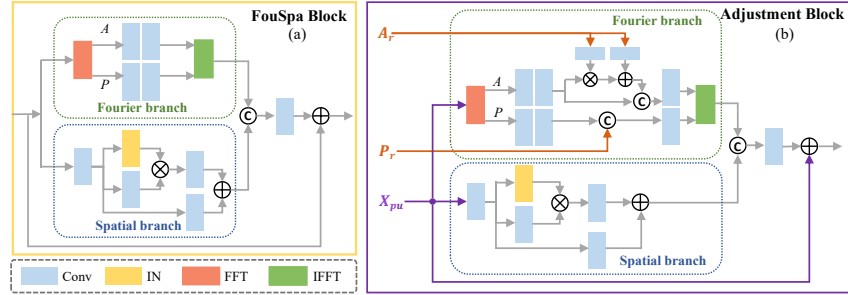

**Figure 4:** Structures of the FouSpa Block (a) and Adjustment Block (b).

**FouSpa Block.** In Sec. 2.1, we observe that luminance and noise can be decomposed in the Fourier domain. Hence, we design the FouSpa Block to parallelly implement amplitude and phase enhancement in the Fourier domain and feature enhancement in the spatial domain. As shown in Figure 4(a), the input features are forked into the Fourier and Spatial branches. In the Fourier branch, FFT is first used to obtain the amplitude component ($A$) and phase component ($P$). The two components are separately fed to two Conv layers with $1 \times 1$ kernel. Note that when processing amplitude and phase, we only use $1 \times 1$ kernel to avoid damaging the structure information. Then, we transform them back to the spatial domain via IFFT and concatenate them with the spatial features enhanced by a Half Instance Normalization (HIN) unit (Chen et al., 2021a). We adopt the HIN unit based on its efficiency. The concatenated features are further fed to a Conv layer and then combined with the input features in a residual manner. Although our main motivations are in the Fourier domain, the use of the spatial branch is necessary. This is because the spatial branch and Fourier branch are complementary. The spatial branch adopts convolution operations that can model the structure dependency well in spatial domain. The Fourier branch can attend global information and benefit the disentanglement of energy and degradation.

**Adjustment Block.** The Adjustment Block is the main structure of the HRNet, and it is lightweight. As shown in Figure 4(b), the Adjustment Block shares a similar structure with the FouSpa Block. Differently, in the Fourier branch, with the refined amplitude $A_r$ features obtained from the LRNet, we use Spatial Feature Transform (SFT) (Wang et al., 2018) to modulate the amplitude features of the input $x_{pu}$ via simple affine transformation. Such a transformation or adjustment is possible because the luminance, as global information, manifests as amplitude components, and the amplitude patterns of an HR scale and its LR scales are similar (as discussed in Sec. 2.1). Note that we cannot modulate the phase because of its periodicity. Besides, we do not find an explicit relationship between the HR scale's phase and its LR scales. However, we empirically find that concatenating the refined phase $P_r$ features achieved from the LRNet with the phase features of the input $x_{pu}$ improves the final performance. We thus apply such concatenation in our solution.

**Losses.** We use $l_1$ to supervise $\hat{y}_8$ and $\hat{y}$. We also add perceptual loss to supervise $\hat{y}_8$ while the use of perceptual loss on $\hat{y}$ is impracticable because of its high resolution. Instead, we add SSIM loss $\mathcal{L}_{ssim}$ on $\hat{y}$. The final loss $\mathcal{L}$ is the combination of these losses:

$$\mathcal{L} = \|\hat{y} - y\|_1 + 0.0004 \times \mathcal{L}_{ssim}(\hat{y}, y) + 0.1 \times \|\hat{y}_8 - y_8\|_1 + 0.0002 \times \|\text{VGG}(\hat{y}_8) - \text{VGG}(y_8)\|_2, \quad (1)$$

where $y$ is the ground truth, $y_8$ is the $8\times$ downsampled version of $y$, VGG is the pre-trained VGG19 network, in which we use four scales to supervise training Zhou et al. (2022).

## 3 UHD-LL DATASET

We collect a real low-noise/normal-clear paired image dataset that contains 2,150 pairs of 4K UHD data saved in 8bit sRGB format. Several samples are shown in Figure 5.

Images are collected from a camera mounted on a tripod to ensure stability. Two cameras, *i.e..*, a Sony $\alpha$7 III camera and a Sony Alpha a6300 camera, are used to offer diversity. The ground truth (or normal-clear) image is captured with a small ISO $\in [100, 800]$ in a bright scene (indoor or outdoor). The corresponding low-noise image is acquired by increasing the ISO $\in [1000, 20000]$ and reducing the exposure time. Due to the constraints of exposure gears in the cameras, shooting in the large ISO range may produce bright images, which opposes the purpose of capturing low-light and noisy

**Table 1:** Comparison between classic LLIE datasets and our UHD-LL dataset. 'Number': the number of paired images. 'Resolution': the average resolution of the dataset. 'Noise': low-light images contain noise. 'Real': both low-light images and GT are acquired in real scenes.

| Dataset | Number | Resolution | Noise | Real |
|---|---|---|---|---|
| SID (RAW) | 5,094 | 4240×2832 6000×4000 | ✓ | ✓ |
| MIT-Adobe FiveK | 5,000 | 4000×2500 | | |
| Exposure-Errors | 24,000 | 1000×900 | | |
| LOL | 500/789 | 600×400 | ✓ | ✓ |
| **UHD-LL (Ours)** | **2,150** | **3840×2160** | ✓ | ✓ |

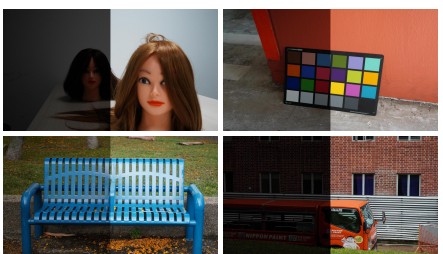

**Figure 5:** Samples from the proposed UHD-LL dataset.

images. Thus, in some cases, we put a neutral-density (ND) filter with different ratios on the camera lens to capture low-noise images. In this way, we can increase the ISO to generate heavier noises and simultaneously obtain extremely dark images, enriching the diversity of darkness and noise levels.

The main challenge of collecting paired data is to reduce misalignment caused by camera shakes and dynamic objects. We take several measures to ameliorate the issue. Apart from using a tripod, we also use remote control software (Imaging Edge) to adjust the exposure time and ISO value to avoid any physical contact with the camera. To further reduce subtle misalignments, we adopt an image alignment algorithm (Evangelidis & Psarakis, 2008) to estimate the affine matrix and align the low-light image and its ground truth. We improve the alignment method by applying AdaIN (Huang & Belongie, 2017) before the affine matrix estimation to reduce the intensity gap. Finally, we hire annotators to check all paired images carefully and discard those that still exhibit misalignments.

We split the UHD-LL dataset into two parts: 2,000 pairs for training and 115 pairs for testing. The training and test partitions are exclusive in their scene and data. We also ensure consistency in pixel intensity distribution between the training and test splits. More analysis of this data, *e.g.*., the pixel intensity and Signal-to-Noise Ratio (SNR) distributions, can be found in the Appendix.

A comparison between our UHD-LL dataset and existing paired low-light image datasets is presented in Table 1. The LOL dataset (two versions: LOL-v1: 500 images; LOL-v2: 789 images) is most related to our UHD-LL dataset as both focus on real low-light images with noise. The LOL-v2 contains all images of the LOL-v1. In contrast to the LOL dataset, our dataset features a more extensive collection, where diverse darkness and noise levels from rich types of scenes are considered. Moreover, the images of our dataset have higher resolutions than those from the LOL dataset. As shown in Figure 1, the models pre-trained on the LOL dataset cannot handle the cases in our UHD-LL dataset due to its insufficient training data, which are low-resolution and contains mostly mild noises. Different from SID dataset that focuses on RAW data, our dataset only studies the data with RGB format. The images in the SID dataset are captured in extremely dark scenes. Its diversity of darkness levels and scenes is limited. When these RAW data with extremely low intensity are transformed into sRGB images, some information would be truncated due to the bit depth constraints of 8bit sRGB image. In this case, it is challenging to train a network for effectively mapping noise and low-light images to clear and normal-light images using these sRGB images as training data.

## 4 EXPERIMENTS

**Implementation.** We implement our method with PyTorch and train it on six NVIDIA Tesla V100 GPUs. We use an ADAM optimizer for network optimization. The learning rate is set to 0.0001. A batch size of 6 is applied. We fix the channels of each Conv layer to 16, except for the Conv layers associated with outputs. We use the Conv layer with stride $= 2$ and $4 \times 4$ kernels to implement the $2\times$ downsample operation in the encoder and interpolation to implement the $2\times$ upsample operation in the decoder in the LRNet. Unless otherwise stated, the Conv layer uses stride $= 1$ and $3 \times 3$ kernels. We use the training data in the UHD-LL dataset to train our model. Images are randomly cropped into patches of size $512 \times 512$ for training.

**Compared Methods.** We include 14 state-of-the-art methods (21 models in total) for our benchmarking study and performance comparison. These methods includes 12 light enhancement methods: NPE (TIP'13) (Wang et al., 2013) SRIE (CVPR'16) (Fu et al., 2016), DRBN (CVPR'20) (Yang et al., 2020a), Zero-DCE (CVPR'20) (Guo et al., 2020), Zero-DCE++ (TPAMI'21) (Li et al., 2021), RUAS (CVPR'21) (Liu et al., 2021b), Zhao *et al.* (ICCV'21) (Zhao et al., 2021), Enlighten-

**Table 2:** Benchmarking study on the testing set of our UHD-LL. All models are released from the original papers and trained on the corresponding datasets. The best and second results are in red and blue, respectively.

| Methods | PSNR↑ | SSIM↑ | LPIPS↓ | MUSIQ↑ | NIQE↓ | NIMA↑ | Training Sets |
|---|---|---|---|---|---|---|---|
| input | 9.926 | 0.482 | 0.551 | 26.779 | 5.379 | 2.269 | - |
| NPE (Wang et al., 2013) | 18.293 | 0.587 | 0.547 | 35.200 | 4.916 | 2.368 | - |
| SRIE (Fu et al., 2016) | 16.316 | 0.652 | 0.503 | 31.345 | 4.927 | 2.110 | - |
| DRBN (Yang et al., 2020a) | 15.455 | 0.689 | 0.450 | 34.925 | 4.408 | 2.154 | LOL-v2 |
| Zero-DCE (Guo et al., 2020) | 17.081 | 0.664 | 0.509 | 35.488 | 5.006 | 2.139 | SICE |
| Zero-DCE++ (Li et al., 2021b) | 17.648 | 0.672 | 0.506 | 32.520 | 4.887 | 2.211 | SICE |
| RUAS-LOL (Liu et al., 2021b) | 11.761 | 0.701 | 0.514 | 28.396 | 5.909 | 2.565 | LOL-v2 |
| RUAS-MIT5K (Liu et al., 2021b) | 14.250 | 0.586 | 0.553 | 29.900 | 5.407 | 2.270 | MIT-Adobe FiveK |
| RUAS-DarkFace (Liu et al., 2021b) | 11.325 | 0.583 | 0.596 | 28.256 | 6.160 | 2.561 | DarkFace |
| Zhao et al.-MIT5K (Zhao et al., 2021) | 15.177 | 0.547 | 0.530 | 32.127 | 4.495 | 2.208 | MIT-Adobe FiveK |
| Zhao et al.-LOL (Zhao et al., 2021) | 18.604 | 0.694 | 0.479 | 32.392 | 4.248 | 2.183 | LOL-v1 |
| EnlightenGAN (Jiang et al., 2021) | 17.637 | 0.767 | 0.459 | 27.441 | 5.497 | 1.977 | Assembled |
| Afifi et al. (Afifi et al., 2021) | 18.212 | 0.610 | 0.479 | 33.970 | 4.793 | 2.217 | Exposure-Errors |
| SCI-easy (Ma et al., 2022) | 15.536 | 0.610 | 0.501 | 31.848 | 4.897 | 2.166 | MIT-Adobe FiveK |
| SCI-medium (Ma et al., 2022) | 15.481 | 0.622 | 0.528 | 31.474 | 4.941 | 2.211 | LOL+LSRW |
| SCI-difficult (Ma et al., 2022) | 17.872 | 0.578 | 0.544 | 36.219 | 5.218 | 2.106 | DarkFace |
| SNR-Aware-LOLv1$_{resize}$ (Xu et al., 2022) | 15.737 | 0.802 | 0.448 | 20.385 | 9.591 | 2.275 | LOL-v1 |
| SNR-Aware-LOLv1$_{stitch}$ (Xu et al., 2022) | 15.536 | 0.695 | 0.468 | 33.098 | 3.961 | 2.387 | LOL-v1 |
| SNR-Aware-LOLv2real$_{resize}$ (Xu et al., 2022) | 15.954 | 0.742 | 0.471 | 23.494 | 9.257 | 2.001 | LOL-v2 |
| SNR-Aware-LOLv2real$_{stitch}$ (Xu et al., 2022) | 14.616 | 0.634 | 0.488 | 33.477 | 4.143 | 2.577 | LOL-v2 |
| SNR-Aware-LOLv2synthetic$_{resize}$ (Xu et al., 2022) | 16.031 | 0.748 | 0.494 | 20.065 | 9.963 | 2.248 | LOL-syn |
| SNR-Aware-LOLv2synthetic$_{stitch}$ (Xu et al., 2022) | 15.887 | 0.675 | 0.497 | 31.473 | 4.460 | 2.484 | LOL-syn |
| URetinex-Net (Wu et al., 2022) | 20.689 | 0.706 | 0.457 | 35.434 | 4.974 | 2.181 | LOL-v1 |

GAN (TIP'21) (Jiang et al., 2021), Afifi et al. (CVPR'21) (Afifi et al., 2021), SCI (CVPR'22) (Ma et al., 2022), SNR-Aware (CVPR'22) (Xu et al., 2022), URetinex-Net (CVPR'22) (Wu et al., 2022) and 2 Transformers: Uformer (CVPR'22) (Wang et al., 2022) and Restormer (CVPR'22) (Zamir et al., 2022). We use their released models and also retrain them using the same training data as our method. Note that some methods provide different models trained using different datasets. Due to the heavy models used in Restormer (Zamir et al., 2022) and SNR-Aware (Xu et al., 2022), we cannot infer the full-resolution results of both methods on UHD images, despite using a GPU with 48G memory. Following previous UHD study (Zheng et al., 2021), we resort to two strategies for this situation: (1) We downsample the input to the largest size that the model can handle and then resize the result to the original resolution, denoted by the subscript 'resize'. (2) We split the input into four patches without overlapping and then stitch the result, denoted by the subscript 'stitch'.

**Evaluation Metrics.** We employ full-reference image quality assessment metrics PSNR, SSIM (Wang et al., 2004), and LPIPS (Alex version) (Zhang et al., 2018) to quantify the performance of different methods. We also adopt the non-reference image quality evaluator (NIQE) (Mittal et al., 2013) and the multi-scale image quality Transformer (MUSIQ) (trained on KonIQ-10k dataset) (Ke et al., 2021) for assessing the restoration quality. We notice that the quantitative results reported by different papers diverge. For a fair comparison, we adopt the commonly-used IQA PyTorch Toolbox[1] to compute the quantitative results of all compared methods. We also test the trainable parameters and running time for processing UHD 4K data.

## 4.1 BENCHMARKING EXISTING MODELS

To validate the performance of existing LLIE methods that were trained using their original training data, we directly use the released models for evaluation on the UHD low-light images. These original training datasets include LOL (Wei et al., 2018), MIT-Adobe-FiveK (Bychkovsky et al., 2011), Exposure-Errors (Afifi et al., 2021), SICE (Cai et al., 2018), LSRW (Hai et al., 2024), and Dark-Face (Yang et al., 2020b). EnlightenGAN uses the assemble training data from existing datasets (Wei et al., 2018; Dang-Nguyen et al., 2015; Kalantari & Ramamoorthi, 2017b; Cai et al., 2018). In addition, the LOL-v1 and LOL-v2 contain real low-light images while LOL-syn is a synthetic dataset. Due to the limited space, we only show relatively good results. As shown in Figure 6, all methods can improve the luminance of the input image. However, they fail to produce visually pleasing results. DRBN and EnlightenGAN introduce artifacts. RUAS-LOL and RUAS-DarkFace yield over-exposed results. Color deviation is observed in the results of EnlightenGAN and Afifi et al. All methods cannot handle the noise well and even amplify noise.

We also summarize the quantitative performance of different methods and verify the effectiveness of commonly used non-reference metrics for UHD low-light images in Table 2. URetinex-Net achieves the highest PSNR score while SNR-Aware-LOLv1 is the best performer in terms of SSIM and

---

[1]https://github.com/chaofengc/IQA-PyTorch

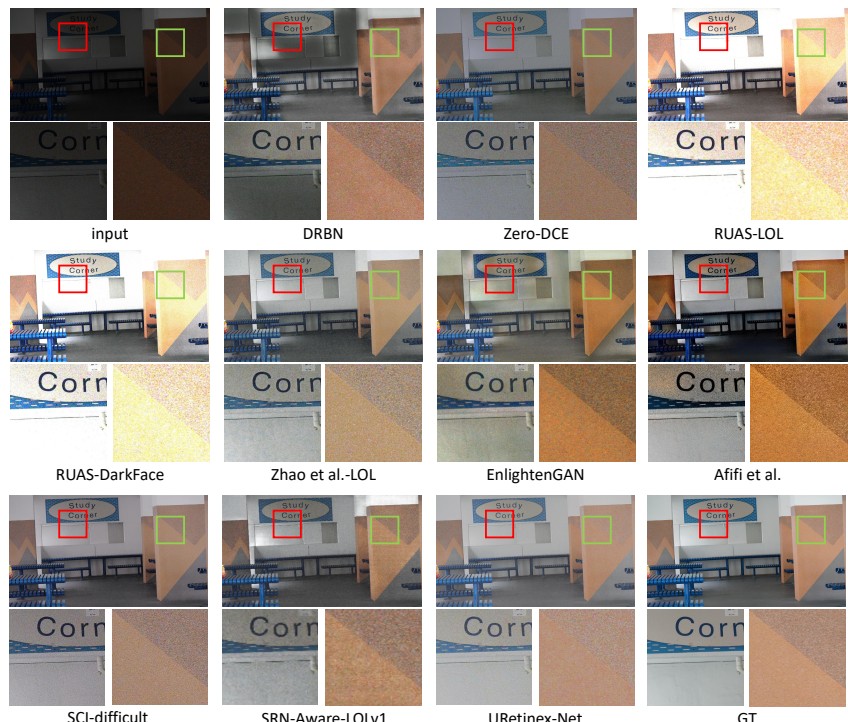

**Figure 6:** Visual comparison between state of the arts for restoring a UHD low-light image. The compared methods include DRBN (Yang et al., 2020a), Zero-DCE (Guo et al., 2020), RUAS-LOL (Liu et al., 2021b), RUAS-DarkFace (Liu et al., 2021b), Zhao et al.-LOL (Zhao et al., 2021), EnlightenGAN (Jiang et al., 2021), Afifi et al. (Afifi et al., 2021), SCI-difficult(Ma et al., 2022), SNR-Aware-LOLv1$_{resize}$ (Xu et al., 2022), and URetinex-Net(Wu et al., 2022). We use the released model directly in this evaluation. All released models cannot handle the UHD low-light image well. More results can be found in the Appendix.

LPIPS. For non-reference metrics, SCI-difficult, Zhao *et al.*-LOL, and RUAS-LOL are the winners under MUSIQ, NIQE, and NIMA, respectively. From Figure 6 and Table 2, we found the non-reference metrics designed for generic image quality assessment cannot accurately assess the subjective quality of the enhanced UHD low-light images. For example, RUAS-LOL suffers from obvious over-exposure in the result while it is the best performer under the NIMA metric.

In summary, the performance of existing released models is unsatisfactory when they are used to enhance the UHD low-light images. The darkness, noise, and artifacts still exist in the results. Compared with luminance enhancement, noise is the more significant challenge for these methods. No method can handle the noise issue well. The joint task of luminance enhancement and noise removal raises a new challenge for LLIE, especially under limited computational resources. We also observe a gap between visual results and the scores of non-reference metrics for UHD LLIE. The gap calls for more specialized non-reference metrics for UHD LLIE.

## 4.2 COMPARING RETRAINED MODELS

Besides the released models, we also retrain existing methods on our UHD-LL training data and compare their performance with our method. Due to the limited space, we only compare our method with several good performers. More results can be found in the Appendix. As shown in Figure 7, our UHDFour produces a clear and normal-light result close to the ground truth. In comparison, Zero-DCE++, RUAS, Afifi *et al.*, SCI, and Restormer experience color deviations. Zero-DCE, Zero-DCE++, RUAS, Zhao *et al.*, Afifi *et al.*, and SCI cannot remove the noise due to the limitations of their network designs. These methods mainly focus on luminance enhancement. SNR-Aware, Uformer, and Restormer have strong modeling capability because of the use of Transformer structures. However, the three methods still leave noise on the results and introduce artifacts.

The quantitative comparison is presented in Table 3. Our UHDFour achieves state-of-the-art performance in terms of PSNR, SSIM, and LPIPS scores and outperforms the compared methods with a

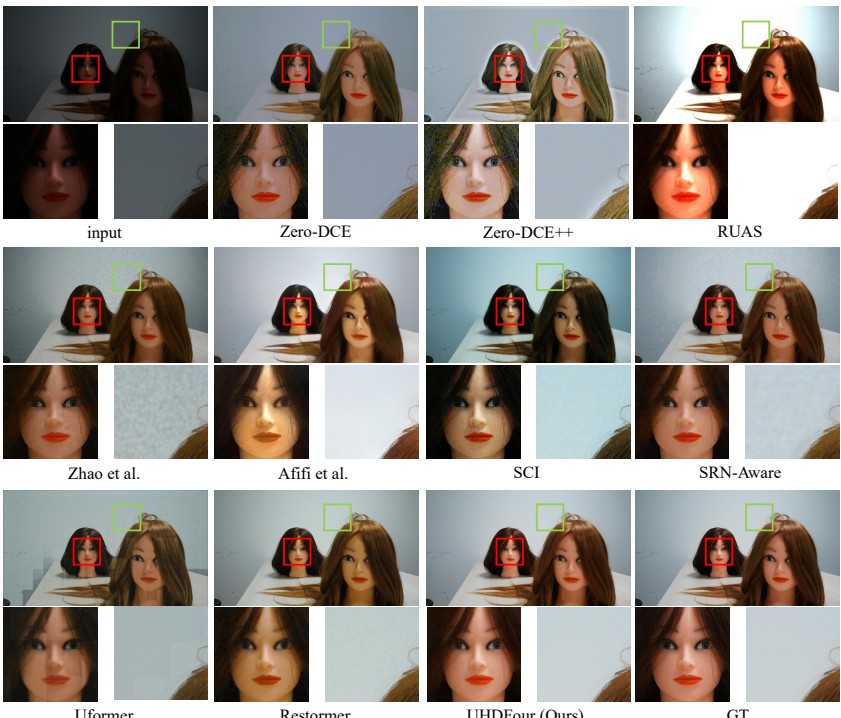

**Figure 7:** Visual comparison between the retrained state of the arts on the UHD-LL dataset. The compared methods include Zero-DCE (Guo et al., 2020), Zero-DCE++ (Li et al., 2021b), RUAS (Liu et al., 2021b), Zhao et al. (Zhao et al., 2021), Afifi et al. (Afifi et al., 2021), SCI (Ma et al., 2022), SNR-Aware (Xu et al., 2022), Uformer (Wang et al., 2022), and Restormer (Zamir et al., 2022). All compared models leave noise, artifacts, or color deviations in the results. Our method achieves a visually pleasing result.

**Table 3:** Quantitative comparison of the retrained state of the arts on the UHD-LL dataset. The best result is in red whereas the second one is in blue. RT: Running time. The training code of URetinex-Net is not released.

| Methods | PSNR↑ | SSIM↑ | LPIPS↓ | Parameter↓ | RT↓ |
|---|---|---|---|---|---|
| Zero-DCE (Guo et al., 2020) | 17.075 | 0.663 | 0.513 | 79.416K | 0.353s |
| Zero-DCE++ (Li et al., 2021b) | 16.410 | 0.630 | 0.530 | 10.561K | 0.327s |
| RUAS (Liu et al., 2021b) | 13.562 | 0.749 | 0.460 | 3.438K | 0.379s |
| Zhao *et al.* (Zhao et al., 2021) | 21.964 | 0.870 | 0.324 | 11.560M | 6.900s |
| Afifi *et al.* (Ma et al., 2020) | 20.805 | 0.740 | 0.440 | 70.154M | 1.631s |
| SCI (Ma et al., 2022) | 16.057 | 0.625 | 0.533 | 0.258K | 0.308s |
| SNR-Aware$_{resize}$ (Xu et al., 2022) | 22.717 | 0.877 | 0.304 | 40.084M | 0.026s |
| SNR-Aware$_{stitch}$ (Xu et al., 2022) | 22.170 | 0.866 | 0.307 | 40.084M | 0.035s |
| Uformer (Wang et al., 2022) | 19.283 | 0.849 | 0.356 | 20.628M | 0.235s |
| Restormer$_{resize}$ (Zamir et al., 2022) | 22.597 | 0.878 | 0.280 | 26.112M | 0.368s |
| Restormer$_{stitch}$ (Zamir et al., 2022) | 22.252 | 0.871 | 0.289 | 26.112M | 0.368s |
| UHDFour (Ours) | 26.226 | 0.900 | 0.239 | 17.537M | 0.024s |

large margin. The Transformer-based SNR-Aware and Restormer rank the second best. Our method has the fastest processing speed for UHD images as most computation is conducted in the LR space.

To further verify the effectiveness of our network, we compare our approach with several methods, including Retinex-Net Wei et al. (2018), Zero-DCE (Guo et al., 2020), AGLL-Net (Lv et al., 2021), Zhao *et al.* (Zhao et al., 2021), RUAS (Liu et al., 2021b), SCI (Ma et al., 2022), and URetinex-Net (Wu et al., 2022), that were pre-trained or fine-tuned on the LOL-v1 and LOL-v2 datasets (Wei et al., 2018). Due to the mild noise and low-resolution images in

**Table 4:** Quantitative comparison on the LOL-v1 and LOL-v2 datasets. The best result is in red whereas the second one is in blue. '-' indicates the pre-trained model is not available.

| Methods | LOL-v1 | | LOL-v2 | |
|---|---|---|---|---|
| | PSNR↑ | SSIM↑ | PSNR↑ | SSIM↑ |
| input | 7.77 | 0.19 | 9.72 | 0.21 |
| Retinex-Net (Wei et al., 2018) | 16.77 | 0.54 | 15.43 | 0.64 |
| Zero-DCE (Guo et al., 2020) | 16.79 | 0.67 | 12.84 | 0.54 |
| AGLLNet (Lv et al., 2021) | 17.52 | 0.77 | 20.69 | 0.78 |
| Zhao *et al.* (Zhao et al., 2021) | 21.67 | 0.87 | 18.84 | 0.84 |
| RUAS (Liu et al., 2021b) | 16.44 | 0.70 | 15.48 | 0.67 |
| SCI (Ma et al., 2022) | 14.78 | 0.62 | 16.74 | 0.62 |
| URetinex-Net (Wu et al., 2022) | 19.84 | 0.87 | - | - |
| UHDFour (Ours) | 23.09 | 0.87 | 21.78 | 0.87 |

the LOL-v1 and LOL-v2 datasets, we change the $8\times$ downsample and upsample operations to $2\times$ and retrain our network. And such characteristics of LOL-v1 and LOL-v2 datasets prohibit us from showing the full potential of our method in removing noise and processing high-resolution images. Even though our goal is not to pursue state-of-the-art performance on the LOL-v1 and LOL-v2 datasets, our method achieves satisfactory performance as presented in Table 4. The visual results are provided in the Appendix.

## 4.3 ABLATION STUDY

We present ablation studies to demonstrate the effectiveness of the main components in our design. For the FouSpa Block, we remove the Fourier branch (FB) (#1), remove the Spatial branch (SB) (#2), and replace the FouSpa Block (*i.e..*, without FB and SB) with the Residual Block of comparable parameters (#3). We also replace the FB with the SB (*i.e..*, using two SB) (#4). For the Adjustment Block, we remove the Amplitude Modulation (AM) (#5), remove the Phase Guidance (PG) (#6), remove the SB (#7), and remove both AM and PG (#8). We also replace the AM and PG with two SB (#9), and

**Table 5:** Quantitative comparison of ablated models. FB: Fourier Branch; SB: Spatial Branch; AM: Amplitude Modulation; PG: Phase Guidance; and Concat: Concatenation. $\times$: multiple repeated modules.

| # | FouSpa Block | | Adjustment Block | | | Output | Performance |
|---|---|---|---|---|---|---|---|
| | FB | SB | AM | PG | SB | Concat | PSNR/SSIM |
| 1 | | ✓ | ✓ | ✓ | ✓ | ✓ | 24.123/0.877 |
| 2 | ✓ | | ✓ | ✓ | ✓ | ✓ | 24.722/0.874 |
| 3 | | | ✓ | ✓ | ✓ | ✓ | 24.005/0.853 |
| 4 | | 2× | ✓ | ✓ | ✓ | ✓ | 24.310/0.878 |
| 5 | ✓ | ✓ | | ✓ | ✓ | ✓ | 25.529/0.883 |
| 6 | ✓ | ✓ | ✓ | | ✓ | ✓ | 24.828/0.874 |
| 7 | ✓ | ✓ | ✓ | ✓ | | ✓ | 24.513/0.872 |
| 8 | ✓ | ✓ | | | ✓ | ✓ | 22.421/0.855 |
| 9 | ✓ | ✓ | | | 3× | ✓ | 24.366/0.867 |
| 10 | ✓ | ✓ | | | | ✓ | 24.106/0.863 |
| 11 | ✓ | ✓ | ✓ | ✓ | ✓ | | 25.616/0.887 |
| 12 | | 2× | | | 3× | ✓ | 23.373/0.851 |
| 13 | ✓ | ✓ | ✓ | ✓ | ✓ | ✓ | 26.226/0.900 |

replace the Adjustment Block with the Residual Block of comparable parameters (#10). For the final output, we remove the concatenation of the LR normal-clear result ($\hat{y}_8$), indicated as #11. We also replace all FB with SB, indicated as #12. Unless otherwise stated, all training settings remain unchanged as the implementation of full model, denoted as #13.

The quantitative comparison of the ablated models on the UHD-LL testing set is presented in Table 5. We also show the visual comparison of some ablated models in the Appendix. As shown, all the key designs contribute to the best performance of the full model. Without the Fourier branch (#1), the quantitative scores significantly drop. The result suggests that processing amplitude and phase separately improves the performance of luminance enhancement and noise removal. From the results of #2, the Spatial branch also boosts the performance. However, replacing the FouSpa Block with the Residual Block (#3) cannot achieve comparable performance with the full model (#13), indicating the effectiveness of the FouSpa Block. For the Adjustment Block, the Amplitude Modulation (#5), Phase Guidance (#6), and Spatial branch (#7) jointly verify its effectiveness. Such a block cannot be replaced by a Residual Block (#10). From the results of #11, we can see that it is necessary to estimate the LR result. In addition, replacing the Fourier branch with spatial branch (#4,#9,#12) cannot achieve comparable performance with the full model (#13), showing the efficacy of Fourier branch.

## 5 CONCLUSION

The success of our method is inspired by the characteristics of real low-light and noisy images in the Fourier domain. Thanks to the unique design of our network that handles luminance and noises in the Fourier domain, it outperforms state-of-the-art methods in UHD LLIE with appealing efficiency. With the contribution of the first real UHD LLIE dataset, it becomes possible to compare existing methods with real UHD low-light images. Our experiments are limited to image enhancement; we have not provided data and benchmarks in the video domain. Our exploration has not considered adversarial losses due to memory constraints. Moreover, as our data is saved in sRGB format, the models trained on our data may fail in processing the extreme cases, in which the information is lost due to the limited bit depth. HDR data may be suitable for these cases. Nevertheless, we believe our method and the dataset can bring new opportunities and challenges to the community. The usefulness of Fourier operations may go beyond our work and see potential in areas like image decomposition and disentanglement. With improved efficiency, it may be adopted for applications that demand real-time response, *e.g..*, enhancing the perception of autonomous vehicles in the dark.

## 6 ETHICS STATEMENT

This study focuses on low-light image enhancement and does not involve any ethics issues. The dataset proposed in this paper also does not involve privacy issues.

## 7 ACKNOWLEDGEMENT

This study is supported under the RIE2020 Industry Alignment Fund Industry Collaboration Projects (IAF-ICP) Funding Initiative, as well as cash and in-kind contribution from the industry partner(s). It is also partially supported by the NTU NAP grant. Chun-Le Guo is also supported by MindSpore, CANN, and Ascend AI Processor.

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

# Appendix

In this Appendix, we present the related work and provide additional results and analysis.

## A  RELATED WORK

**Low-Light Image Enhancement Methods.** The focus of our work is on deep learning-based LLIE methods (Li et al., 2021a; Liu et al., 2021a). Wang et al. (2019) proposed a network to enhance the underexposed photos by estimating an image-to-illumination mapping. EnlightenGAN (Jiang et al., 2021) proposed an attention-guided network to make the generated results indistinguishable from real normal-light images. By formulating light enhancement as a task of image-specific curve estimation that can be trained with non-reference losses, Zero-DCE (Guo et al., 2020) and Zero-DCE++ (Li et al., 2021b) obtain good brightness enhancement. Zhao et al. (2021) treated underexposed image enhancement as image feature transformation between the underexposed image and its paired enhanced version. Liu et al. (2021b) proposed a Retinex model-inspired unrolling method, in which the network structure is obtained by neural architecture search. Afifi et al. (2021) proposed a coarse-to-fine network for exposure correction. Ma et al. (2022) proposed a self-calibrated illumination learning framework using unsupervised losses. Wu et al. (2022) combined the Retinex model with a deep unfolding network, which unfolds an optimization problem into a learnable network. Xu et al. (2022) proposed to exploit the Signal-to-Noise Ratio (SNR)-aware Transformer and convolutional models for LLIE. In this method, long-range attention is used for the low SNR regions while the short-range attention (convolutional layers) is for other regions.

Different from these works, our network takes the challenging joint luminance enhancement, noise removal, and high resolution constraint of UHD low-light images into account in the Fourier domain, endowing new insights on UHD LLIE and achieving better performance.

**Low-Light Image Enhancement Datasets.** LOL (Wei et al., 2018) dataset contains pairs of low-/normal-light images saved in RGB format, in which the low-light images are collected by changing the exposure time and ISO. Due to the small size, it only covers a small fraction of the noise and darkness levels. MIT-Adobe FiveK (Bychkovsky et al., 2011) dataset includes paired low-/high-quality images, where the high-quality images are retouched by five experts. The low-quality images are treated as low-light images in some LLIE methods. However, this dataset is originally collected for global tone adjustment, and thus, it ignores noise in its collection. Based on the MIT-Adobe FiveK dataset, a multi-exposure dataset, Exposure-Errors, is rendered to emulate a wide range of exposure errors (Afifi et al., 2021). Similar to the MIT-Adobe FiveK dataset, the Exposure-Errors dataset also neglects the noise issue. SID (Chen et al., 2018) is a RAW data dataset.

The images of the SID dataset have two different sensor patterns (*i.e.*, Bayer pattern and APS-C X-Trans pattern). Due to the specific data pattern, the deep models trained on this dataset are not versatile as they require the Raw data with the same pattern as input. Besides, a long-exposure reference image corresponds to multiple short-exposure images, leading to limited scene diversity.

Unlike existing LLIE datasets that either omit noise, capture limited numbers of images, require specific sensor patterns, or exclude UHD images, we propose a real UHD LLIE dataset that contains low-noise/normal-clear image pairs with diverse darkness and noise levels captured in different scenarios. A comprehensive comparison is presented in Sec. 3.

**Image Decomposition-based Enhancement.** There are some image decomposition-based enhancement methods. For LLIE, Xu et al. (2020) proposed a frequency-based decomposition-and-enhancement model, which suppresses noises in the low-frequency layer and enhances the details in the high-frequency layer. Yang et al. (2020a) proposed a band representation-based semi-supervised model. This method consists of two stages: recursive band learning and band recomposition. Wei et al. (2018) also decomposed a low-light image into an illumination component and a reflectance component according to the Retinex model and then separately enhance the components. Image decomposition was also used in shadow removal, in which two networks are used to predict shadow parameters and matter layer (Le & Samaras, 2019).

In addition, assuming that phase preserves high-level semantics while the amplitude contains low-level features, (Guo et al., 2022) proposed a FPNet that consists of two stages for image de-raining. The first state is to restore the amplitude of rainy images. The second state then refines the phase of

the restored rainy images. To solve the limitations of ResBlock that may overlook the low-frequency information and fails to model the long-distance information, (Mao et al., 2021) proposed a Residual Fast Fourier Transform with Convolution Block for image deblurring. The Block contains a spatial residual stream and a FFT stream. (Pham et al., 2021) proposed a complex valued neural network with Fourier transform for image denoising. The complex valued network first converts the noisy image to complex value via Fourier transform, then estimates a complex filter which is applied to the converted noisy image for approaching the complex value of the ground truth image.

Although these methods decompose an image or use Fourier transform in the networks, our design has different motivations. Our motivations are inspired by the uniqueness of the Fourier domain for UHD low-light image enhancement as presented in Figure 2, i.e., luminance and noise can be 'decomposed' to a certain extent in the Fourier domain and HR image and its LR versions share similar amplitude patterns. We also design the specific blocks to use these characteristics and solve the high-resolution constraint issue, which have not been explored in previous works.

**HDR Imaging.** There are some high dynamic range (HDR) reconstruction works (Salih et al., 2012; Wang & Yoon, 2021) that are related to our low-light image enhancement. The HDR reconstruction also can be grouped into multi-image HDR reconstruction and single-image reconstruction. The multi-image methods require to fuse the multiple bracketed exposure low dynamic range (LDR) images. To mitigate artifacts caused by image fusion, several technologies (Srikantha & Sidibe, 2012) have been proposed. For single-image methods, deep learning has achieved impressive performance. In addition to learning end-to-end LDR-to-HDR networks (Kalantari & Ramamoorthi, 2017a; Wu et al., 2018; Yang et al., 2018; Zhang & Lalonde, 2017; Hu et al., 2022b; Eilertsen et al., 2017) some methods either synthesize multiple LDR images with different exposures (Endo et al., 2017) or model the inverse process of the image formation pipeline (Liu et al., 2020). Besides, some works also focus on HDR reconstruction and denoising. For example, (Hu et al., 2022a) proposes a joint multi-scale denoising and tone-mapping framework, which prevents the tone-mapping operator from overwhelming the denoising operator. (Chen et al., 2021b) takes the noise and quantization into consideration for designing the HDR reconstruction network.

# B   ANALYSIS OF UHD-LL DATASET

We first show the intensity histograms and the SNR distribution of our UHD-LL dataset in Figure 8. The SNR is computed using the same algorithm as the recent LLIE method (Xu et al., 2022). As shown in Figure 8(a) and Figure 8(b), when splitting the training and testing sets, we make the pixel intensity distributions of the training and testing sets consistent to guarantee the rationality of the dataset split. We also plot the SNR distribution of the dataset to show the noise levels in Figure 8(c). The SNR distribution suggests the wide and challenging SNR ranges of our dataset. We show more samples and amplify the resolution of our UHD-LL data in Figure 9.

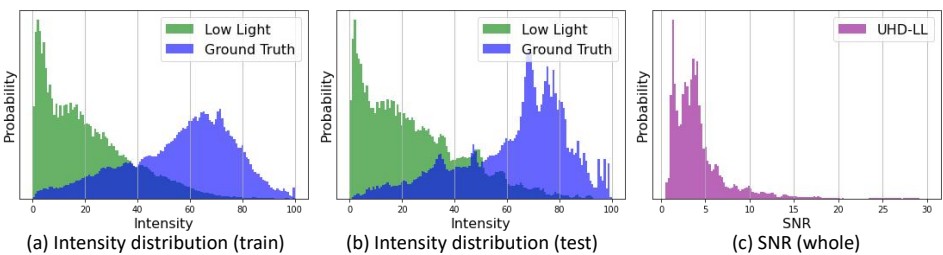

**Figure 8:** Distribution analysis of the UHD-LL dataset. Pixel intensity histograms of images for low-noise (green) and normal-clear (blue) from the training and test partitions are shown in (a) and (b), suggesting their consistent pixel intensity distributions. (c) The whole dataset covers a wide range of SNR distribution ranging from 1 to 30 and the SNR values center at the range of [0,10], indicating the challenging noise levels.

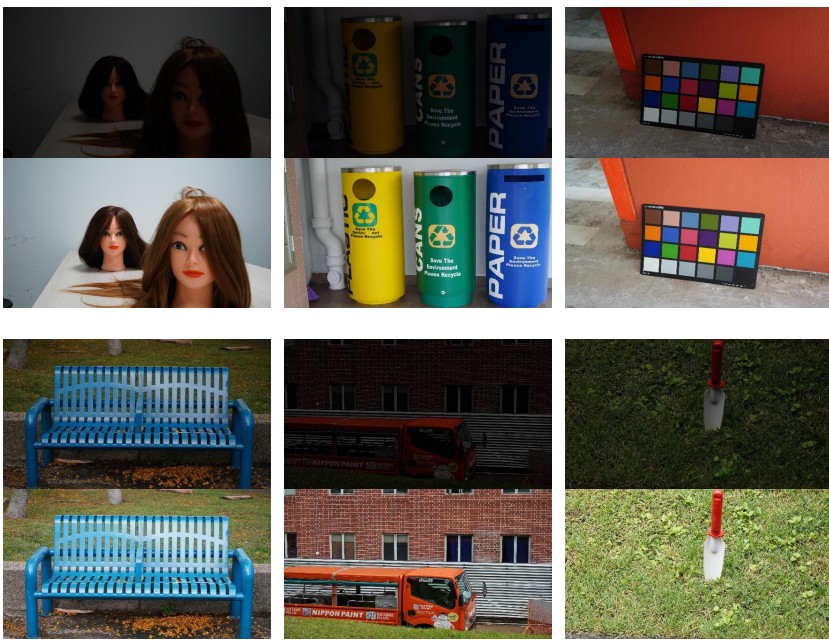

**Figure 9:** Samples of the UHD-LL dataset. The UHD-LL dataset contains 2,150 pairs of 4K UHD low-noise/normal-clear data, covering real noises, diverse darkness levels, and a large variety of scenes.

## C   FURTHER ANALYSIS OF MOTIVATION

Recall that in Sec. 2.1, we discussed two observations that serve as the motivation to design our network. In particular, (a) Swapping the amplitude of a low-noise image with that of its corresponding normal-clear image produces a normal-noise image and a low-clear image, and (b) The amplitude patterns of an HR normal-clear image and its LR versions are similar and are different from the corresponding HR low-noise counterpart.

We first show more motivation cases in Figures 10, 11, and 12. These visual results suggest the same tendency as the motivations shown in the main paper.

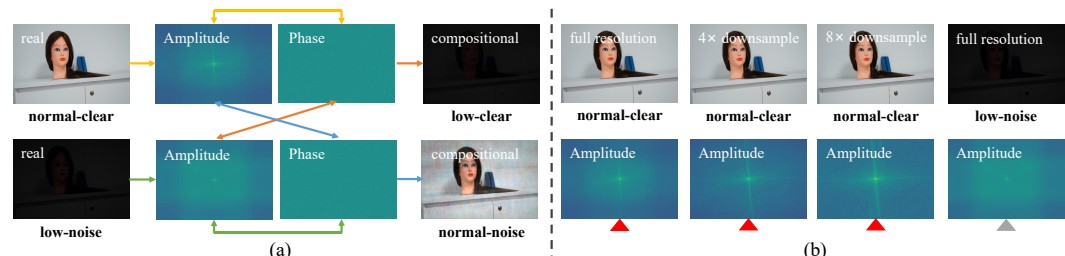

**Figure 10:** Examples of our motivations.

To further analyze our first motivation, we compare the luminance and noise of real normal-clear and low-noise images and compositional low-clear and normal-noise images. To compare the luminance similarity, we compute the average luminance. For real noise level measurement, there is no corresponding metric. Thus, we use the recent multi-scale Transformer MUSIQ (Ke et al., 2021) for image quality assessment. MUSIQ is not sensitive to luminance changes. Moreover, it can be used to measure the noise level as its training dataset contains noisy data and it shows state-of-the-art performance for assessing the quality of natural images. A large MUSIQ value reflects better image quality with less noise and artifacts. We select 50 images from the UHD-LL dataset and compare the average scores. We present the quantitative results in Table 6.

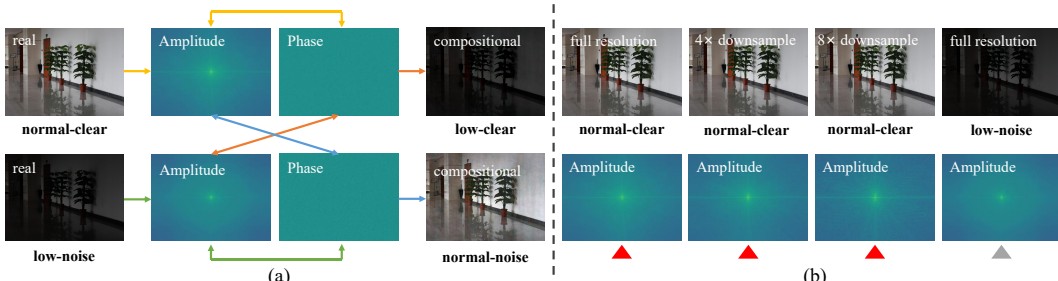

**Figure 11:** Examples of our motivations.

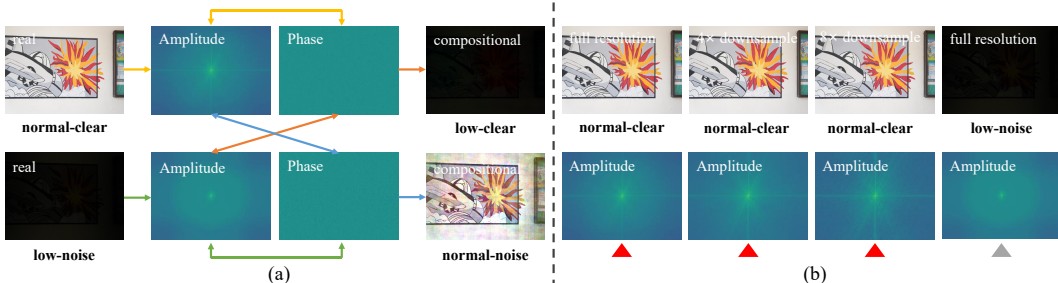

**Figure 12:** Examples of our motivations.

**Table 6:** Quantitative results to support our motivation (a). The measurement metrics are the average luminance value/MUSIQ value.

| | Real | | Compositional | |
|---|---|---|---|---|
| | normal-clear | low-noise | low-clear | normal-noise |
| UHD-LL dataset | 147.25/61.22 | 20.50/32.75 | 23.25/55.50 | 125.00/33.55 |

As presented in Table 6, the real normal-clear images have similar luminance values with the compositional normal-noise images while they have similar high MUSIQ values with the compositional low-clear images. Similarly, the real low-noise images have similar luminance values with the compositional low-clear images while they have similar low MUSIQ values with the compositional normal-noise images. The results further suggest that luminance and noise can be decomposed to a certain extent in the Fourier domain. Specifically, luminance would manifest as amplitude while noise is closely related to phase.

For the second motivation, it is difficult to quantify the similarity of amplitude spectrum of different sizes as it cannot be directly interpolated. The full-reference metrics cannot be used in this situation. Hence, we show more visual examples in Figure 10(b), Figure 11(b), and Figure 12(b). The extra examples support our motivation.

## D    VISUALIZATION IN THE NETWORK

We show the changes of amplitude and phase in our proposed UHDFour network in Figure 13. As shown, the amplitude and phase of our final result are similar with those of ground truth. Moreover, the amplitude and phase of the low-resolution output $\hat{y}_8$ are also similar with those of its corresponding ground truth $y_8$. We wish to emphasize that although noise is related to phase, it cannot be explicitly represented in phase imagery format as phase represents the initial position of the wave. Only the combination of amplitude and phase can express a complete image. Moreover, in the feature domain, such relevance is more difficult to be represented in an imagery format. Thus, we suggest to see the similarity between the final result and ground truth, instead of the intermediate features and phase.

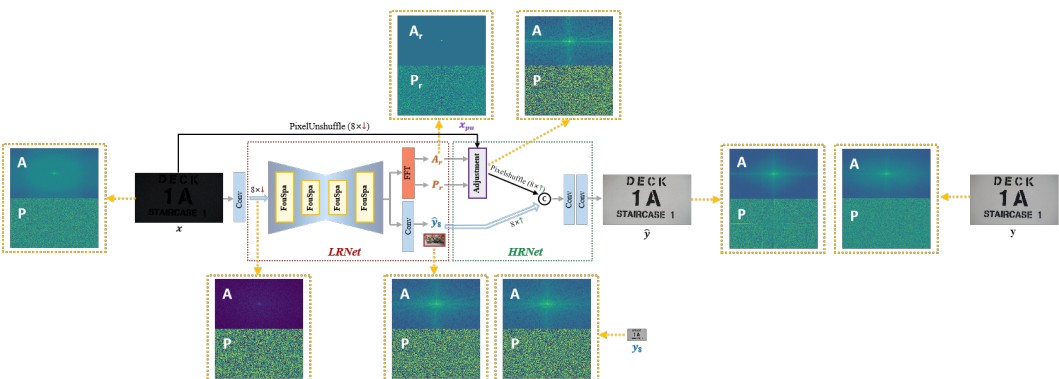

**Figure 13:** Visualization of the amplitude and phase changes in the proposed UHDFour network. We show the amplitude and phase components in yellow boxes. For visualization, we show the amplitude and phase in imagery format with common transformations. Note that phase is periodic and cannot be accurately represented in imagery format. $y$ is the corresponding ground truth. $y_8$ is the $8\times$ downsampled ground truth.

# E  MORE RESULTS ON RELEASED MODELS

We present more comparisons between state of the arts for restoring UHD low-light images in Figures 14, 15, and 16. This is similar to Figure 6 of the main paper where we compare methods using their original released models. As shown, all existing models cannot handle the UHD low-light images well. Since we cannot infer the full-resolution results of SNR-Aware Xu et al. (2022) on UHD images, despite using a GPU with 48G memory, some obvious borders appear in its results due to the stitching strategy. The phenomenon also indicates the commonly-used stitching strategy in previous UHD data processing methods is inapplicable to the challenging UHD low-light image enhancement task.

# F  MORE RESULTS ON RETRAINED MODELS ON UHD-LL

We provide more visual comparisons of our method with retrained state-of-the-art methods on the UHD-LL dataset in Figures 17 and 18.

As the results shown, for UHD low-light image enhancement, the retrained models on the UHD-LL dataset still cannot achieve satisfactory results. Noise and artifacts can still be found in their results. The results suggest that joint luminance enhancement and noise removal in the spatial domain is difficult. Our solution effectively handles this challenging problem by embedding Fourier transform in a cascaded network, in which luminance and noise can be decomposed to a certain extent and are processed separately.

# G  MORE RESULTS ON RETRAINED MODELS ON LOL-V1 AND LOL-V2 DATASETS

We also provide more visual comparisons of our method with the models that were pre-trained or fine-tuned on the LOL-v1 and LOL-v2 datasets in Figures 19, 20, and 21.

As for the low-light images in LOL-v1 and LOL-v2 datasets, even though the mild noise and low-resolution images prohibit us from showing the full potential of our method in removing noise and processing high-resolution images, our method still achieves satisfactory performance. The results suggest the potential of our solution in different circumstances.

# H  ABLATION STUDY

We show some visual results of the ablated models in Figure 22. Without the Fourier branch (#1), the ablated model cannot effectively enhance luminance and remove noise. Although the result of

**Table 7:** The computational and time costs of FFT/IFFT operations for processing different scales of features. We compute FLOPs (in M) and running time (in second).

| Scales | FFT | | IFFT | |
|---|---|---|---|---|
| | FLOPs | Running time | FLOPs | Running time |
| $8\times \downarrow$ | 35.22 | $2.62\times10^{-4}$ | 35.22 | $5.46\times10^{-4}$ |
| $16\times \downarrow$ | 7.77 | $8.10\times10^{-5}$ | 7.77 | $1.39\times10^{-4}$ |
| $32\times \downarrow$ | 1.68 | $6.34\times10^{-5}$ | 1.68 | $1.17\times10^{-4}$ |

#2 looks better than #1, the Spatial branch (#2) also affects the final result. Directly replacing the FouSpa Block with the Residual Block of comparable parameters (#3) cannot obtain a satisfactory result, suggesting the good performance of the FouSpa Block is not because of the use of more parameters. Removing the Amplitude Modulation (#5) results in the visually unpleasing result. The Phase Guidance (#6) and Spatial branch (#7) in the Adjustment Block also contribute to the good performance of the full model. Directly replacing the Adjustment Block with the Residual Block of comparable parameters (#10) still cannot obtain a satisfactory result. The introduction of the estimation of low-resolution result (#11) also leads to a clear result. The visual comparisons further show the significance of the proposed FouSpa Block and Adjustment Block in dealing with the intricate issue of joint luminance enhancement and noise removal while remaining efficient.

Additionally, we list the computational and time costs of FFT/IFFT operations for processing different scales of features. In our network, we fix the feature channels to 16 and use three different scales $8\times$, $16\times$, and $32\times$ downsampled original resolution (*i.e.*, $3840 \times 2160$). The results are presented in Table 7. The FFT and IFFT operations have same computational cost. The difference in running time may be because of the different optimization strategies used in PyTorch.

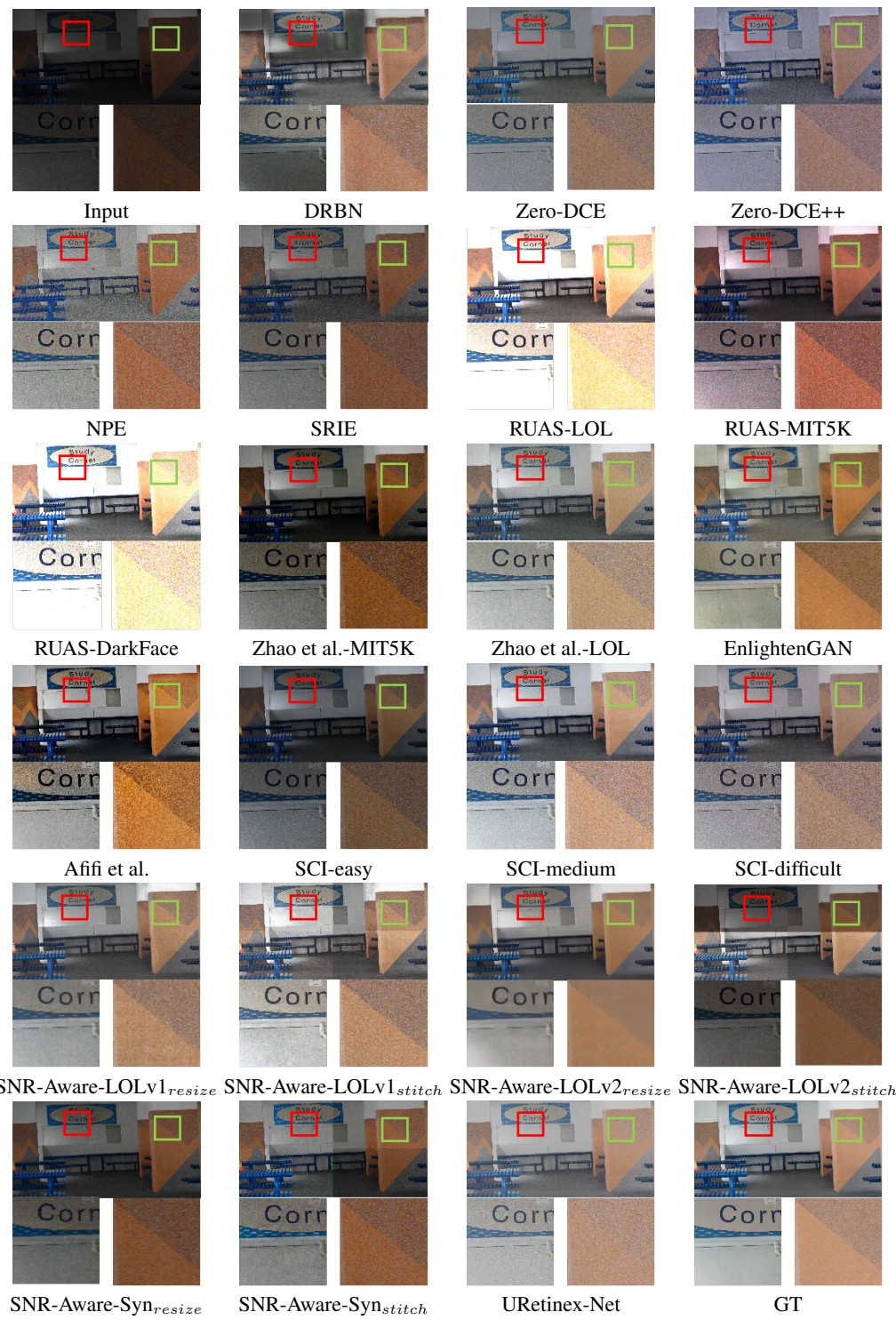

**Figure 14:** Visual comparison of the released state of the arts for restoring a UHD low-light image. The compared methods include DRBN (Yang et al., 2020a), Zero-DCE (Guo et al., 2020), Zero-DCE++ (Li et al., 2021b), NPE (Wang et al., 2013), SRIE (Fu et al., 2016), RUAS-LOL (Liu et al., 2021b), RUAS-MIT5K (Liu et al., 2021b), RUAS-DarkFace (Liu et al., 2021b), Zhao et al.-MIT5K (Zhao et al., 2021), Zhao et al.-LOL (Zhao et al., 2021), EnlightenGAN (Jiang et al., 2021), Afifi et al. (Afifi et al., 2021), SCI-easy (Ma et al., 2022), SCI-medium (Ma et al., 2022), SCI-difficult(Ma et al., 2022), SNR-Aware-LOLv1$_{resize}$ (Xu et al., 2022), SNR-Aware-LOLv1$_{stitch}$ (Xu et al., 2022), SNR-Aware-LOLv2$_{resize}$ (Xu et al., 2022), SNR-Aware-LOLv2$_{stitch}$ (Xu et al., 2022), SNR-Aware-Syn$_{resize}$ (Xu et al., 2022), SNR-Aware-Syn$_{stitch}$ (Xu et al., 2022), and URetinex-Net(Wu et al., 2022).

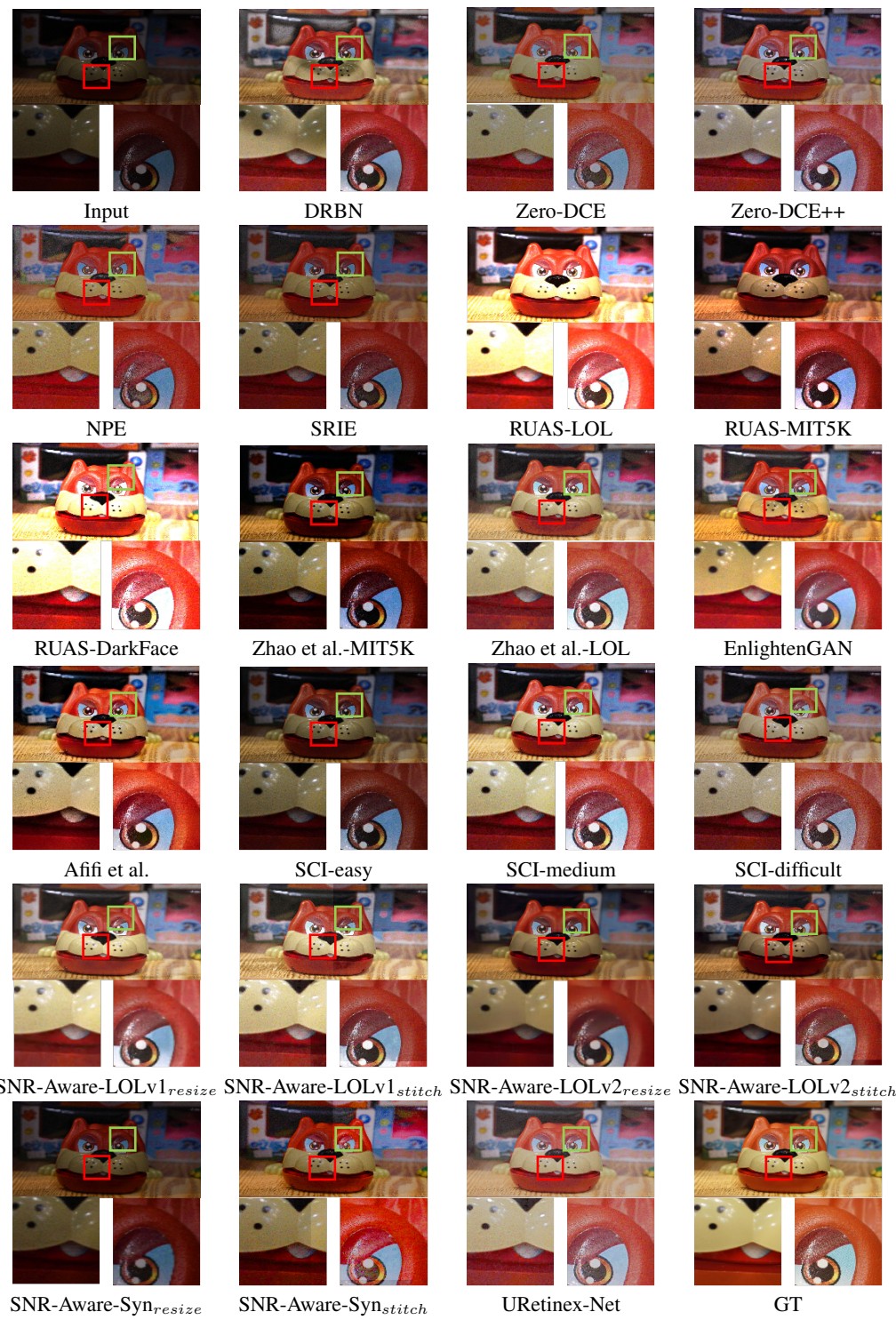

**Figure 15:** Visual comparison of the released state of the arts for restoring a UHD low-light image. The compared methods include DRBN (Yang et al., 2020a), Zero-DCE (Guo et al., 2020), Zero-DCE++ (Li et al., 2021b), NPE (Wang et al., 2013), SRIE (Fu et al., 2016), RUAS-LOL (Liu et al., 2021b), RUAS-MIT5K (Liu et al., 2021b), RUAS-DarkFace (Liu et al., 2021b), Zhao et al.-MIT5K (Zhao et al., 2021), Zhao et al.-LOL (Zhao et al., 2021), EnlightenGAN (Jiang et al., 2021), Afifi et al. (Afifi et al., 2021), SCI-easy (Ma et al., 2022), SCI-medium (Ma et al., 2022), SCI-difficult(Ma et al., 2022), SNR-Aware-LOLv1$_{resize}$ (Xu et al., 2022), SNR-Aware-LOLv1$_{stitch}$ (Xu et al., 2022), SNR-Aware-LOLv2$_{resize}$ (Xu et al., 2022), SNR-Aware-LOLv2$_{stitch}$ (Xu et al., 2022), SNR-Aware-Syn$_{resize}$ (Xu et al., 2022), SNR-Aware-Syn$_{stitch}$ (Xu et al., 2022), and URetinex-Net(Wu et al., 2022).

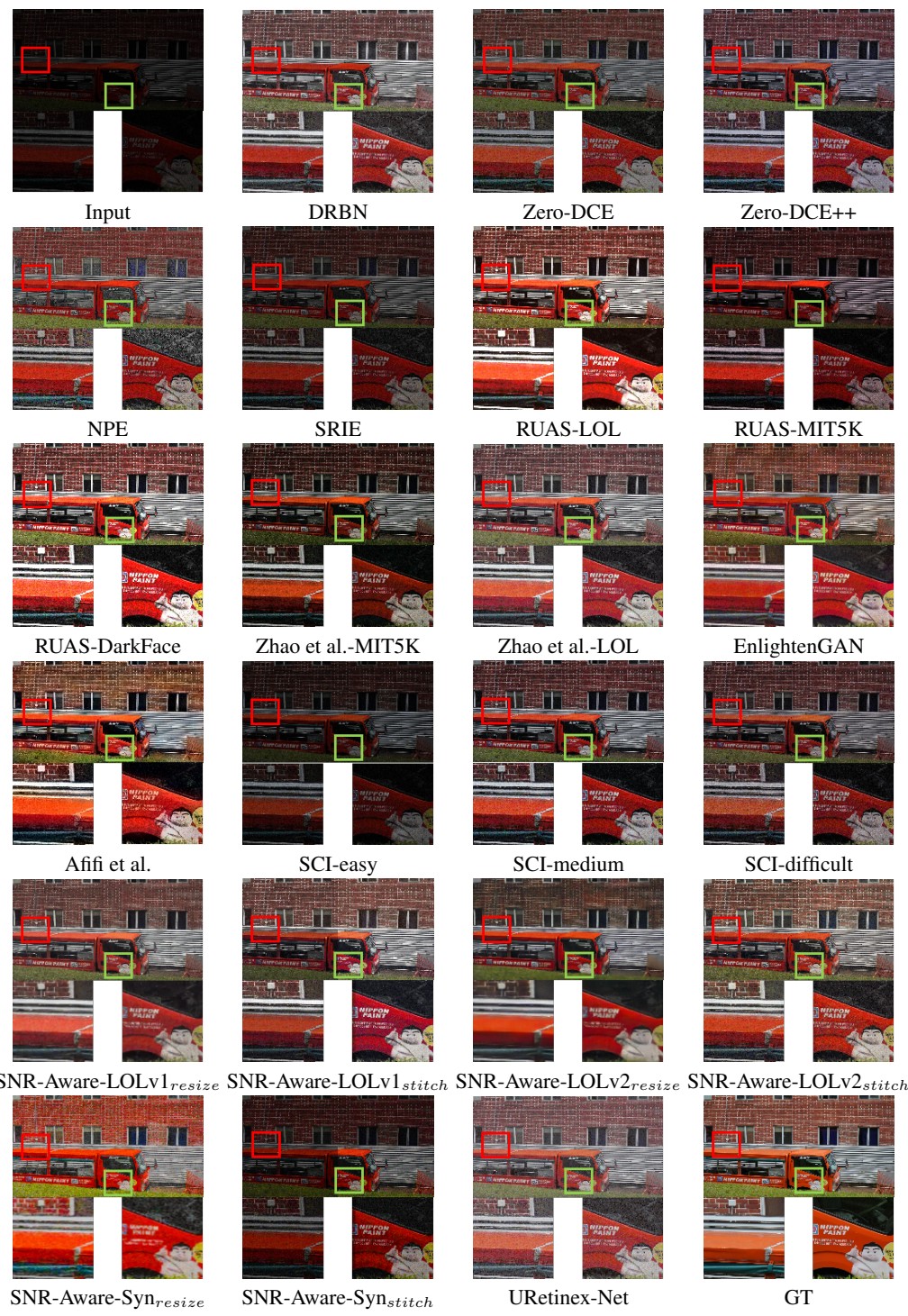

**Figure 16:** Visual comparison of the released state of the arts for restoring a UHD low-light image. The compared methods include DRBN (Yang et al., 2020a), Zero-DCE (Guo et al., 2020), Zero-DCE++ (Li et al., 2021b), NPE (Wang et al., 2013), SRIE (Fu et al., 2016), RUAS-LOL (Liu et al., 2021b), RUAS-MIT5K (Liu et al., 2021b), RUAS-DarkFace (Liu et al., 2021b), Zhao et al.-MIT5K (Zhao et al., 2021), Zhao et al.-LOL (Zhao et al., 2021), EnlightenGAN (Jiang et al., 2021), Afifi et al. (Afifi et al., 2021), SCI-easy (Ma et al., 2022), SCI-medium (Ma et al., 2022), SCI-difficult(Ma et al., 2022), SNR-Aware-LOLv1$_{resize}$ (Xu et al., 2022), SNR-Aware-LOLv1$_{stitch}$ (Xu et al., 2022), SNR-Aware-LOLv2$_{resize}$ (Xu et al., 2022), SNR-Aware-LOLv2$_{stitch}$ (Xu et al., 2022), SNR-Aware-Syn$_{resize}$ (Xu et al., 2022), SNR-Aware-Syn$_{stitch}$ (Xu et al., 2022), and URetinex-Net(Wu et al., 2022).

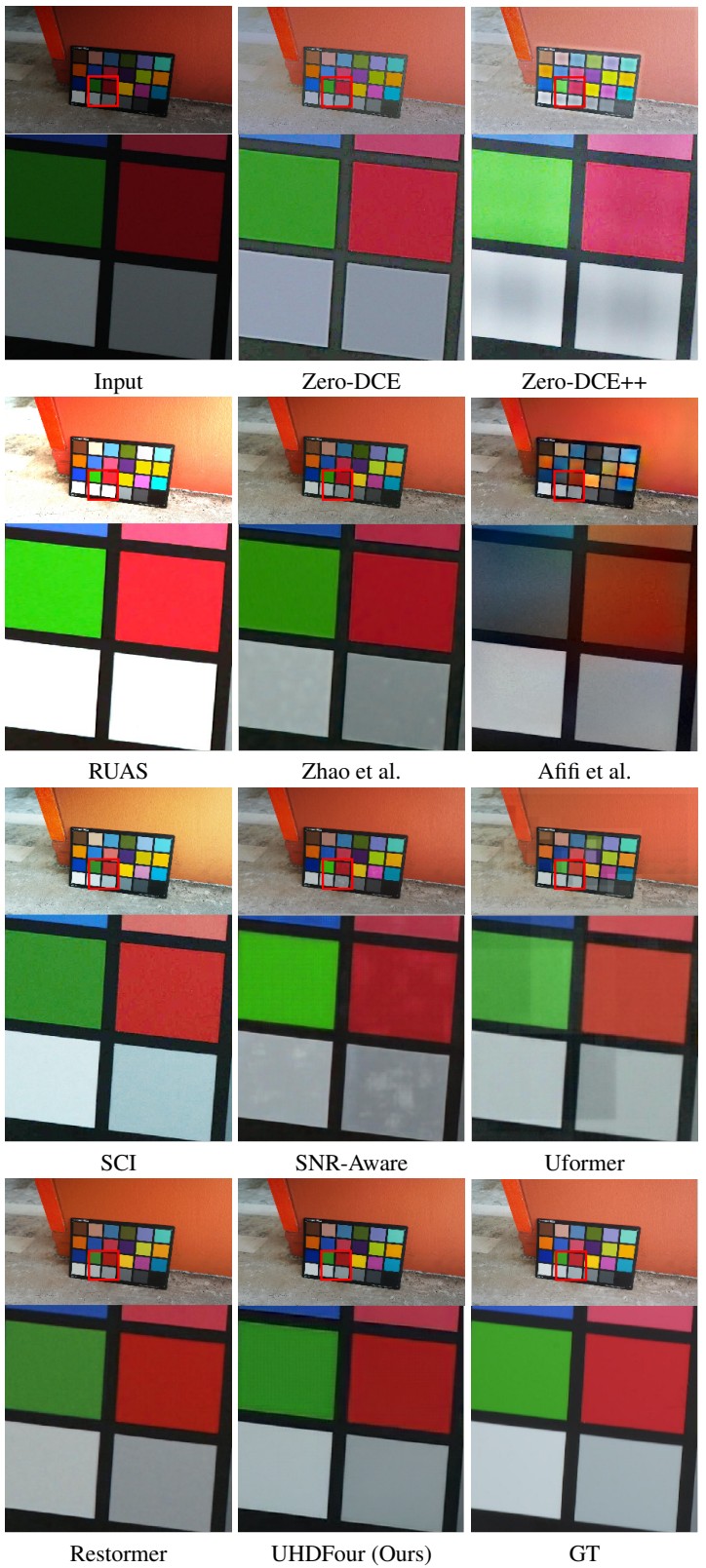

**Figure 17:** Visual comparison of the retrained state of the arts on the UHD-LL dataset. The compared methods include Zero-DCE (Guo et al., 2020), Zero-DCE++ (Li et al., 2021b), RUAS (Liu et al., 2021b), Zhao et al. (Zhao et al., 2021), Afifi et al. (Afifi et al., 2021), SCI (Ma et al., 2022), SNR-Aware (Xu et al., 2022), Uformer (Wang et al., 2022), and Restormer (Zamir et al., 2022).

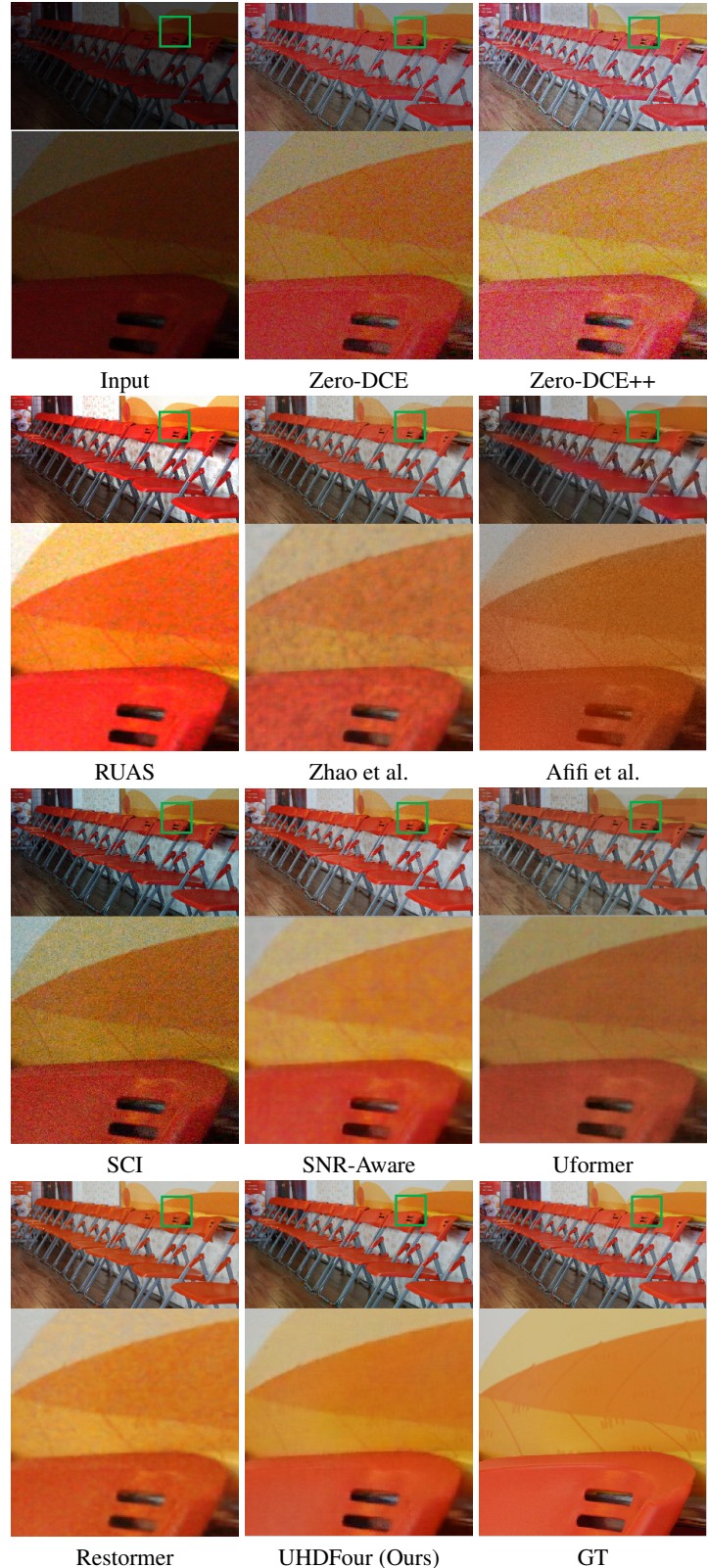

**Figure 18:** Visual comparison of the retrained state of the arts on the UHD-LL dataset. The compared methods include Zero-DCE (Guo et al., 2020), Zero-DCE++ (Li et al., 2021b), RUAS (Liu et al., 2021b), Zhao et al. (Zhao et al., 2021), Afifi et al. (Afifi et al., 2021), SCI (Ma et al., 2022), SNR-Aware (Xu et al., 2022), Uformer (Wang et al., 2022), and Restormer (Zamir et al., 2022).

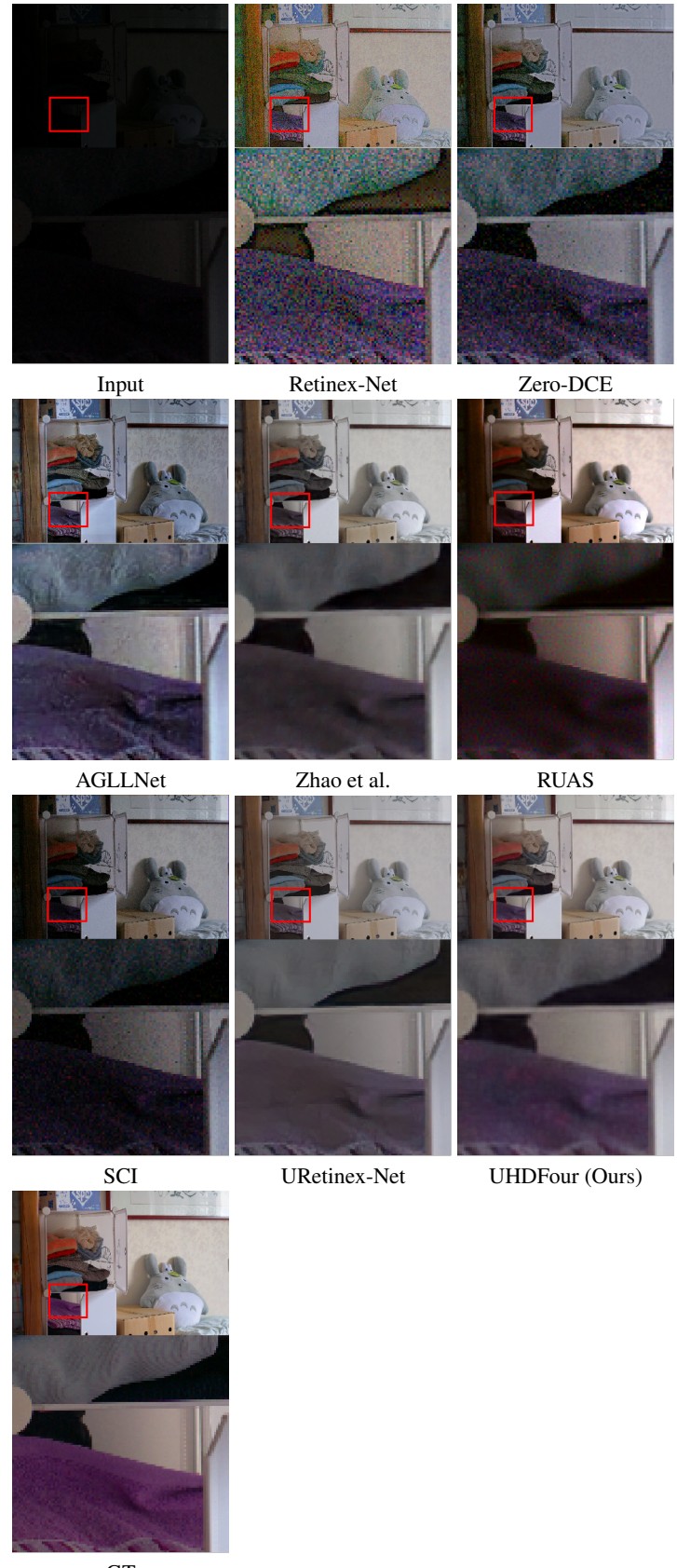

**Figure 19:** Visual comparison on the LOL-v1 dataset. The compared methods include Retinex-Net Wei et al. (2018), Zero-DCE (Guo et al., 2020), AGLLNet Lv et al. (2021), Zhao et al. (Zhao et al., 2021), RUAS (Liu et al., 2021b), and URetinex-Net Wu et al. (2022)

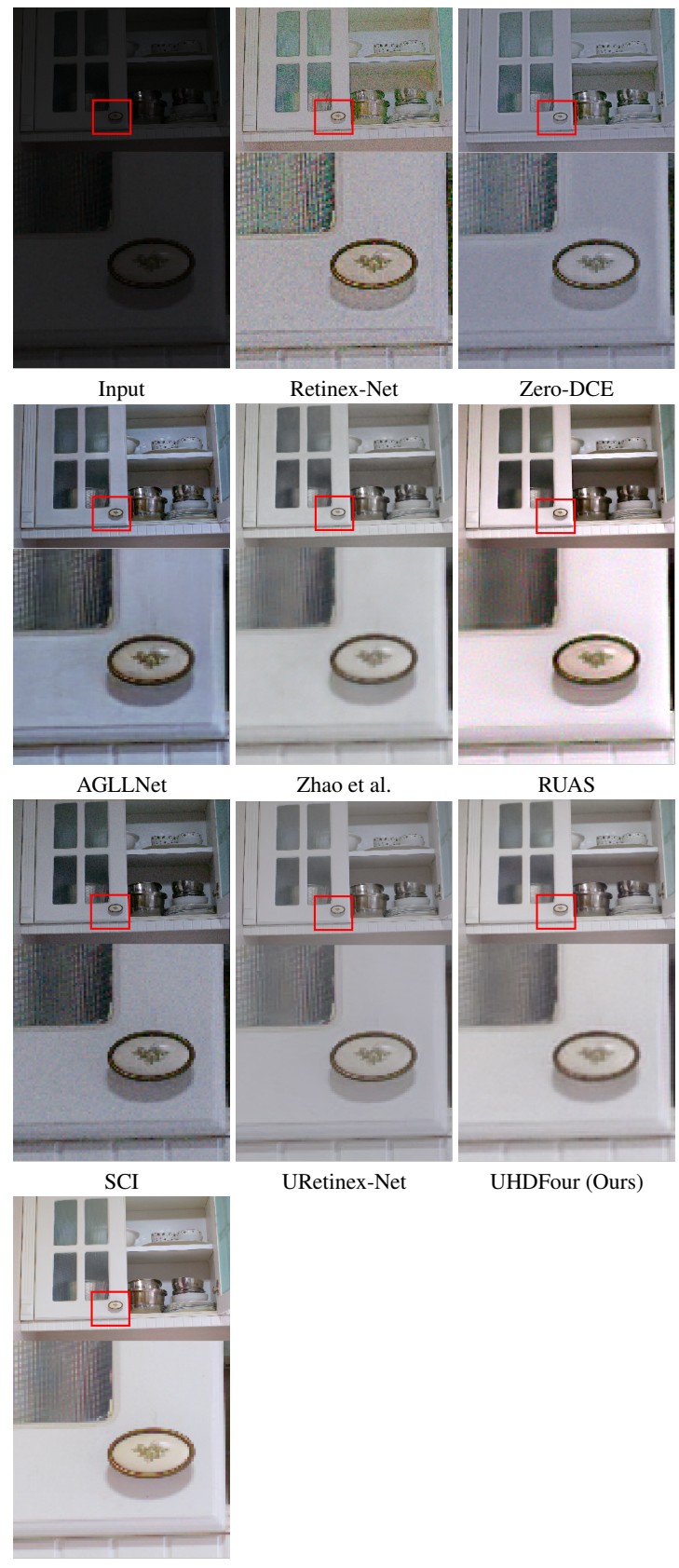

**Figure 20:** Visual comparison on the LOL-v1 dataset. The compared methods include Retinex-Net Wei et al. (2018), Zero-DCE (Guo et al., 2020), AGLLNet Lv et al. (2021), Zhao et al. (Zhao et al., 2021), RUAS (Liu et al., 2021b), and URetinex-Net Wu et al. (2022)

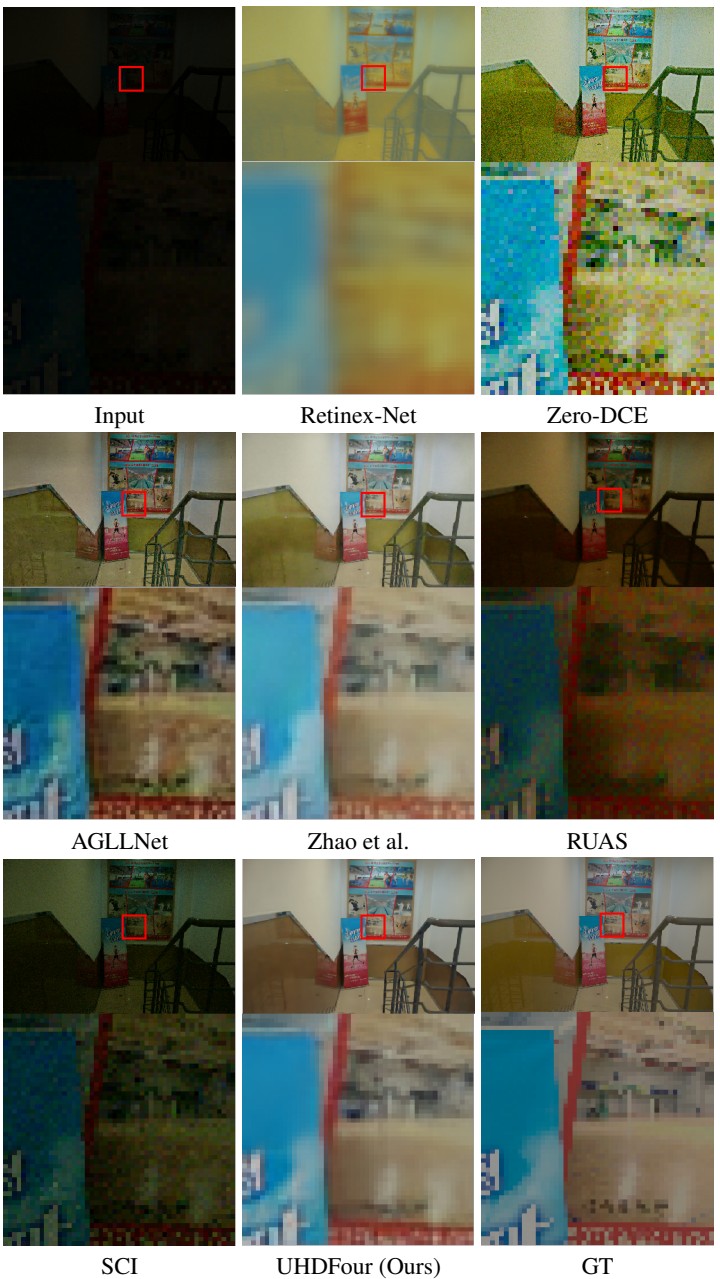

**Figure 21:** Visual comparison on the LOL-v2 dataset. The compared methods include Retinex-Net Wei et al. (2018), Zero-DCE (Guo et al., 2020), AGLLNet Lv et al. (2021), Zhao et al. (Zhao et al., 2021), and RUAS (Liu et al., 2021b)

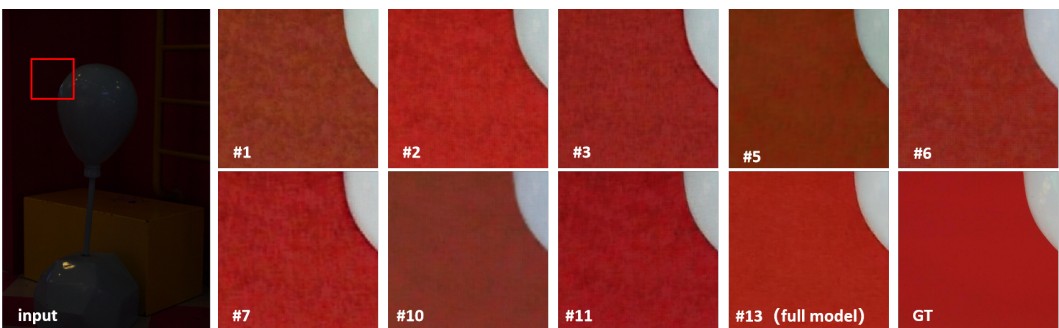

**Figure 22:** Visual results of ablated models.

