# OpenReview forum: "Embedding Fourier for Ultra-High-Definition Low-Light Image Enhancement"
_ICLR.cc/2023/Conference — ICLR 2023 notable top 5%_

### Official Review · Reviewer_aXaB · 2022-10-21

**Confidence:** 5
**Correctness:** 3
**Technical Novelty And Significance:** 3
**Empirical Novelty And Significance:** 3
**Recommendation:** 8

**Clarity, Quality, Novelty And Reproducibility:**

The paper is generally written well and easy to read. The paper shows some interesting observations, based on which it presents a new method for ultra-high-resolution image enhancement. A new dataset is constructed and will be released to the public together with the code, as the authors promise.

**Strength And Weaknesses:**

Strong points.
1. A new method is proposed for ultra high-resolution low-light noisy image enhancement with notable performance advantages against existing methods.
2. The network design has a clear motivation with some interesting experiments to justify the observations.
3. A new ultra high-resolution dataset of 2150 low-noise/normal-clear paired 4K images is constructed.
4. The paper is written well and easy to follow.
5. Code and dataset will be released.

Weak points.
1. The first observation is justified via visual illustrations and quantitative experiments while the second one is illustrated by visual comparisons. It would be better if the authors can provide some interpretations. Can authors explain from the FFT why these phenomena happen? The second observation only mentions that HR images and their corresponding LR images share similar amplitude patterns, but how about the phases? Do they have similar patterns? If not, would the noise removal in LR version affect that in HR images?

2. In the 3rd paragraph, Introduction, there are some vague descriptions when talking about existing methods, e.g., "existing methods are mainly..." and "some studies...". I suggest adding references here so that readers can have direct references.

3. The LRNet/FouSpa block designs seem not to be aligned with the observations. For the FouSpa block, it is mentioned that "we observe that luminance and noise can be decomposed in the Fourier domain", but the design (Fig. 4) shows that the Fourier branch only separately processes the A and P, then combines them together using IFFT. There is still a Spatial branch that works with the Fourier branch. It is not clear why there should still have a spatial branch and how the Fourier branch enhances luminance in A and removes noise in P. Regarding the LRNet design (Fig. 3), it is not clear why FFT and Conv layers are paralleled. Further, do A_r and P_r have any supervision signals? Would the authors show some intermediate visualizations of A_r and P_r?

4. Regarding the dataset acquisition, it seems that images are taken in normal-light scenes, and low-light images are captured with under-exposures. In this case, it seems the noise would not be very severe (e.g., in Figure 5 it is hard to see noise). This makes this paper more related to under-exposure image enhancement but not low-light image enhancement. It would be better if the authors could provide more justifications for this issue. The dataset split is 2000 for training and 115 pairs for test (The train/test ratio is 20:1), which I think is extreme. It would be better if the authors could justify this.

5. A minor issue is that "Table 5" on page 8 should be "Table 4".

6. In Section A of the Appendix, I think it would be better if the authors can discuss how FFT is used in other low-level vision tasks, e.g., image deraining, denoising, and deblurring, and highlight the differences to the proposed method.

**Summary Of The Paper:**

This paper proposes a new method (UHDFour) for enhancing ultra high-resolution low-light noisy images. The proposed method is based on two observations. The first observation is that most luminance information exists on amplitudes while noise exists on phases of the Fourier domain. This observation drives the proposed UHDFour method to process amplitudes and phases separately for luminance enhancement and noise removal. The second observation is that a high-resolution image and its corresponding low-resolution version have similar amplitude patterns, which results in a cascaded network design that first processes a low-resolution image and then processes a high-resolution one. The authors construct a new ultra high-resolution image dataset (UHD-LL). The authors conduct experiments to justify the motivations and the effectiveness of proposed method.

**Summary Of The Review:**

I am on the positive side of this paper but with some questions listed in the above section. I am willing to raise my rating if the authors can address them.

---

> ### Author Response · Authors · 2022-11-17
> **Response to Reviewer aXaB (Part 1)**
>
> We sincerely thank you for reviewing our paper and providing us with valuable feedback. We have addressed your concerns below. The revised parts have been marked in blue in the updated paper and Appendix.
>
> >**Can authors explain from the FFT why these phenomena happen?**
>
> The nature of FFT for an image is to express the image in the discrete frequency domain. In the frequency domain, any frequency spectrums can be expressed by different amplitudes and phases with several fixed frequencies. Among, the amplitude represents the distribution of energy in the frequency domain. The phase represents the initial position of the wave, that is, the way the wave is arranged. In terms of our two main observations, we explain them from the view of FFT as follows.
>
> (1) Observation 1: most luminance information concentrates on amplitudes while noise is closely related to phases. For an image, its low frequencies dominate main energy that is related to amplitude. Moreover, luminance changes in an image mainly affect low frequencies, thus affecting amplitude. In terms of phases, they indicate the major "appearance" e.g., structure, texture, and content of the image. Thus, most noises are in the phase. However, the noises in the phase lack intensity information as phase only represents  the initial position of the wave.
>
> (2) Observation 2: a high-resolution image and its low-resolution version share similar amplitude patterns. According to the Nyquist sampling theorem, when sampling an image at a low sampling rate (e.g., image downsampling), the frequency components are aliased at high frequencies while the low frequencies remain constant. The low frequencies of an image dominate the main energy, showing great values in the center of an amplitude pattern map (the center represents low frequencies). Therefore, the amplitudes (related to low frequencies) of an HR image and its downsampled version are similar because of the low frequency invariance. Enhancing the amplitude of the LR image helps the enhancement of the amplitude of the corresponding HR image as they have similar patterns. The distribution of phase (phase represents the initial position of the wave) manifests not only in low frequencies. Thus, the phase of the LR image does not have similar patterns to that of the HR image. However, retaining the phase of the LR image is significant as only the combination of amplitude and phase can express a complete image. Otherwise, only using amplitude lacks the structure and content information of the image.
>
> >**In the 3rd paragraph, there are some vague descriptions when talking about existing methods. I suggest adding references here.**
>
> Thank you for your suggestion. As suggested, we have added the corresponding references for the descriptions in the third paragraph of the updated paper.

---

> > ### Author Response · Authors · 2022-11-17
> > **Response to Reviewer aXaB (Part 2)**
> >
> > >**For the FouSpa block, it is not clear why these should still have a spatial branch and how the Fourier branch enhances luminance in A and removes noise in P. Regarding the LRNet design, it is not clear why FFT and Conv layers are paralleled. Further, do A_r and P_r have any supervision signals? Would the authors show some intermedia visualizations of A_r and P_r?**
> >
> > The reasons why we still use a spatial branch are explained as follows.  Although our main motivations are in the Fourier domain, the use of the spatial branch is necessary. The spatial branch and Fourier branch are complementary. The spatial branch adopts convolution operations that can model the structure dependency well in spatial domain. The Fourier branch can attend global information and benefit the disentanglement of energy and degradation. Thus, we introduce a spatial branch in our FouSpa block. We have explained the reason on page 4 of the updated paper.
> >
> > The Fourier branch is effective as the ablation study indicated in Section 4.3 of the paper. Removing the Fourier branch, the performance significantly drops. We show the changes of A and P after going through the network in Figure 13 of the updated Appendix. We wish to emphasize that although noise is related to phase, such relevance cannot be explicitly represented in phase imagery format as phase is periodic and represents the initial position of the wave. Only  the combination of amplitude and phase can express a complete image. Moreover, in the feature domain, such relevance is more difficult to be represented in a phase imagery format. Even though, we provide the visualizations for your consideration.
> >
> > In the LRNet, the FFT and Conv layers are paralleled as we add the supervision for the estimation of LR result in the spatial domain. The LR result $y_8$ needs to be produced by a Conv layer in the LRNet. The use of middle supervision in the LRNet is aligned with network optimization as we use the pixel-level losses in the spatial domain to optimize the network. At the same time, the purpose of paralleled FFT is to feed the features of amplitude and phase to the Adjustment block that adjusts the amplitude and phase of HR input. The ablation studies show the guidance information from both the Fourier domain and spatial domain in the LRNet are useful. Besides, the ablation study also shows it is necessary to estimate the LR result  $y_8$.
> >
> > We do not use any supervision signals for A_r and P_r. We optimize them using the final loss function. The visualizations of A_r and P_r are shown in Figure 13 of the updated Appendix.
> >
> >
> >
> > >**It seems the noise would not be very severe. The dataset split is 2000 for training and 115 pairs for test (The train/test ratio is 20:1), which I think is extreme.**
> >
> > As presented on page 4 of the manuscript, the low-noise images are acquired by increasing the ISO in the range of [1000,20000] and reducing the exposure time. Due to the constraints of exposure gears in the cameras, shooting in the large ISO range may produce bright images, which opposes the purpose of capturing low-light and noisy images. Thus, in some cases, we put a neutral-density (ND) filter with different ratios on the camera lens to capture low-noise images.
> >
> > As we know, the ISO in such ranges, i.e., [1000,20000] would produce severe noise. We also plot the SNR distribution of the dataset to show the noise levels in Figure 8(c) of the Appendix. The SNR distribution also suggests the challenging noise levels of our dataset. There are more obvious examples of  severe noise in the paper, such as the two UHD low-light images of Figure 1 and the input of Figure 6. To show the noise, we amplify the brightness of the input UHD low-light images 10 times in Figure 1.
> >
> > For the task of low-light image enhancement, our split is common. For example, for the commonly used LOL-v1 dataset, the training data is 475 while the testing data is 15 (32:1). In our dataset, we carefully split the training set and testing set to guarantee the rationality of the dataset split. As shown in Figure 8(a) and (b) of the Appendix, when splitting the training and test sets, we preserve the pixel intensity distributions of training and testing sets consistent.
> >
> >
> >
> > >**A minor issue is that “Table 5” on page 8 should be “Table 4”.**
> >
> > Thank you for carefully reading. We have revised it as Table 4.
> >
> >
> >
> > >**In Section A of the Appendix, I think it would be better if the authors can discuss how FFT is used in other low-level tasks and highlight the differences to the proposed method.**
> >
> > Thank you for your kind suggestion. We have introduced how FFT was used in the related works and discussed the differences in Section A of the updated Appendix.

---

> > > ### Comment · Reviewer_aXaB · 2022-11-24
> > > **Post-rebuttal**
> > >
> > > Thanks for the response. It addresses my concerns.

---

### Official Review · Reviewer_yq2M · 2022-10-22

**Confidence:** 5
**Correctness:** 3
**Technical Novelty And Significance:** 3
**Empirical Novelty And Significance:** 4
**Recommendation:** 8

**Clarity, Quality, Novelty And Reproducibility:**

The paper is of high quality and provides interesting and important insights into UHD low-light image enhancement. The paper is well-written and easy to follow. The proposed method in the paper is original and novel. Besides, the paper presents the first real UHD low-light image enhancement dataset.

**Strength And Weaknesses:**

Strengths:

+ New task. UHD low-light image enhancement task is becoming more and more important, especially on mobile devices. The reviewer is happy to see there is a paper focusing on this issue and proposing a new solution and the first dataset.
+ Novel solution. Facing challenging issues such as joint luminance enhancement and noise removal under the ultra-high-resolution constraint, this paper proposes an interesting and insightful solution. Different from previous spatial methods, this method tasks full use of the unique characteristics of Fourier domain for this task. In the Fourier domain, the amplitude and phase of a low-light image can be separately processed to avoid amplifying noise when enhancing luminance. Moreover, the proposed method can be scalable to UHD images by implementing amplitude and phase enhancement under the low-resolution regime and then adjusting the high-resolution scale with few computations. The solution is novel and interesting.
+ New dataset. This paper contributes the first UHD low-light image enhancement dataset with paired data. The dataset not only provides sufficient and diverse data but also presents some techniques which show how to minimize geometric and photometric misalignment when capturing the paired low-light/normal-light data. The data collection details provide some insights into the pair data collection.
+ Systematical analysis. This paper systematically analyzes the performance of existing low-light image enhancement methods for processing UHD images. The baselines cover almost all state-of-the-art methods, and the experiments include quantitative and quantitative results. From the results, it is clear to see the advantages and disadvantages of previous methods.
+ Outstanding performance. The proposed method in the paper achieves state-of-the-art performance for UHD low-light image enhancement with the fastest inference speed. Besides, the proposed method can be slightly modified and then achieves state-of-the-art performance on existing low-light image enhancement dataset despite it is not designed for data of this sort.
+ Sufficient experiment and analysis. The paper provides sufficient comparison experiments to show the advantages of the proposed method in the main paper and appendix. In addition, the ablation studies and motivation analysis are convincing.

Weaknesses:

+ The paper analyzes the performance of existing low-light image enhancement for processing UHD low-light images. The baselines are all deep learning-based methods. The reviewer wonders whether traditional methods still cannot solve this challenging issue either. It would be good if some traditional methods could be included for comparison in the final version.
+ For the Sec. 4.1 Benchmarking existing models, the quantitative scores in terms of non-reference metrics could be removed when the paired data is available. It is almost common that non-reference metrics are not much reliable, especially for low-light images.
+ In Figures 6-8 and Tables 2-4, for each method, the corresponding reference should be provided to improve the readability of the paper. It is the same for the figures in the appendix.
+ The reviewer wonders whether the main idea in Fourier domain can be extended to other UHD image related low-level tasks.

**Summary Of The Paper:**

This paper focuses on the Ultra-High-Definition (UHD) low-light image enhancement, which is a new task. Different from previous low-light image enhancement, the task presented in the paper faces more challenging and practical issues such as dealing with the intricate issue of joint luminance enhancement and noise removal while remaining efficient. The paper first shows the limitations of previous methods and datasets for processing the UHD low-light images.  Then, the paper proposes to embed Fourier transform into a cascaded network to address these issues. The designs are interesting and insightful, which are based on the unique characteristics in the Fourier domain. In comparison to spatial domain, these characteristics of Fourier domain make the task easy to deal with. Besides these designs, the paper also presents the first real UHD low-light image enhancement with paired data. The paper also provides some techniques about how to capture the dataset of this kind. The proposed solution achieves the best performance on UHD low-light image enhancement and fast inference speed. Also, the solution can be extended and used for processing existing low-light image dataset.

**Summary Of The Review:**

The paper presents a novel solution and a new dataset. Moreover, the analysis and motivations of this work provide new insights into this challenging task. The experiments are sufficient, and the results are convincing. These are the important factors in my rating.

---

> ### Author Response · Authors · 2022-11-17
> **Response to Reviewer yq2M**
>
> We sincerely thank you for reviewing our paper and providing us with valuable feedback. We have addressed your concerns below. The revised parts have been marked in blue in the updated manuscript and Appendix.
>
> >**It would be good if some traditional methods could be included for comparison in the final version.**
>
>  As suggested, we have added two classical traditional methods, SRIE and NPE, for comparison in the updated paper. The quantitative results are added in Table 2 of the updated paper. We also show their visual results in Figures 14, 15, and 16 of the updated Appendix.
>
>
>
> >**The quantitative scores in terms of non-reference metrics could be removed.**
>
> Thank you for the suggestion. We keep the scores of non-reference metrics to support our arguments that non-reference metrics designed for generic image quality assessment cannot accurately assess the subjective quality of the enhanced UHD low-light images. The gap calls for more specialized non-reference metrics for UHD LLIE.
>
>
>
> >**The corresponding reference should be provided in figures and tables.**
>
>  As suggested, we have provided the corresponding reference for each method in figures and tables of the updated paper and Appendix.
>
>
> >**The reviewer wonders whether the main idea in Fourier domain can be extended to other UHD image related low-level tasks.**
>
> The main idea has the potential to be applied to other UHD image related low-level visual tasks.
>
> From our previous experiments, in terms of image dehazing and image de-raining tasks, we found that the haze/rain layer can be mainly expressed as amplitudes in the Fourier space. Moreover, the amplitude patterns of a UHD ground truth image and its low-resolution versions are similar and different from the corresponding degraded UHD input image.

---

### Official Review · Reviewer_Fqm7 · 2022-10-23

**Confidence:** 3
**Correctness:** 3
**Technical Novelty And Significance:** 3
**Empirical Novelty And Significance:** 3
**Recommendation:** 8

**Clarity, Quality, Novelty And Reproducibility:**

I think the novelty of this paper is enough and the paper is complete and easy to understand.

**Strength And Weaknesses:**

Strength:
1. This paper is well written and easy to understand.
2. The performance is great.
3. The novelty is enough for ICLR. The feature modeling in frequency domain and the dataset.

Weakness.
1. I'm little confused about table 2. Why the performance of proposed method is not shown in Table 2?
2. I suggest authors also show the Amplitude and Phase in the network. People can see how the network improve the Amplitude and Phase.

**Summary Of The Paper:**

This paper propsoe the UHDFour solution for UHD enhancement. This paper introduce a new dataset for 4K UHD, called UHD LLIE. Also this paper designed a module utilize the frequency domain.

**Summary Of The Review:**

I think this paper is complete and has enough novelty.

---

> ### Author Response · Authors · 2022-11-17
> **Response to Reviewer Fqm7**
>
> We sincerely thank you for reviewing our paper and providing us with valuable feedback. We have addressed your concerns below.  The revised parts have been marked in blue in the updated paper and Appendix.
>
> >**I’m little confused about table 2. (Why the performance of proposed method is not shown in Table 2?)**
>
> Thank you for your question. In Table 2, we mainly validate the performance of existing low-light image enhancement models that were trained using their original training data. We also list their training sets in Table 2. As claimed in the paper (on page 7), by analyzing Table 2, we show that the performance of existing released models is unsatisfactory when they are used to enhance the UHD low-light images.
>
> Together with other models that are retrained using the same training data provided by the UHD-LL dataset, the performance of the proposed method is shown in Table 3.
>
>
>
> >**I suggest authors also show the Amplitude and Phase in the network. (People can see how the network improve the Amplitude and Phase.)**
>
> Thank you for your comments. As suggested, we show the changes of A and P after going through the network in Figure 13 of the updated Appendix. We wish to emphasize that although noise is related to phase, such relevance cannot be explicitly represented in phase imagery format as phase is periodic and represents the initial position of the wave. Only  the combination of amplitude and phase can express a complete image. Moreover, in the feature domain, such relevance is more difficult to be represented in a phase imagery format. Even though, we provide the visualizations for your consideration.

---

### Official Review · Reviewer_ohoL · 2022-10-25

**Confidence:** 5
**Correctness:** 3
**Technical Novelty And Significance:** 3
**Empirical Novelty And Significance:** 3
**Recommendation:** 6

**Clarity, Quality, Novelty And Reproducibility:**

The paper is well-written and easy to follow. The quality is up to ICLR's bar. The method, to my knowledge, is novel, and the datasets would be a nice addition to the research community in low-light imaging. Detailed network layer description or table, or source code instead, should be provided, otherwise, reproducing the paper's results can be difficult given the complexity of the proposed network architecture and various comparison and ablation studies performed.

**Strength And Weaknesses:**

### Strength
1. A reasonably sized, high-resolution new dataset for joint luminance enhancement and denoising is provided.
1. The proposed network architecture cleverly mixes and matches the amplitude and phase of the low light input and normal light GT to constrain the denoising network.
### Weakness
1. The importance of the Fourier branch (FB) compared to a more conventional spatial branch (SB) in FourSpa is less than expected and should be further validated. Swapping FB with SB brings about a $0.6~dB$ difference in PSNR, as ablation studies #1 and #2 show, which made me wonder if 2xSB (i.e. increase the size of the SB) would have similar performance as FB+SB in FourSpa. Moreover, for the adj. block, it's necessary to show the performance of SB only and SB increase to the comparable size of SB+AM+PG. Lastly, there should be a Fourier-free architecture where there are only SBs (of the equivalent size of SB + FB + AM + PG + SB, e.g. #9) in both FourSpa and Adj. Block. The author should consider adding those in the final version.
1. What are the computational and time costs of FFT/IFFT-related operations? It will be useful if the author can add this to the ablation study table.
1. Even for consumer audio/video market, "HDR" and "4K" are oftentimes mentioned together. Moreover, HDR is common for low-light scenes as light sources can often be seen in night-time photography. However, HDR scenes are not considered in this paper. I suggest the author mention such drawbacks of the proposed dataset in discussion and/or indication of the dynamic range of the proposed datasets (i.e. the specs of the cameras used for the data capture).
1. Also, since an important aspect of this paper is luminance enhancement, it's natural to discuss HDR tone-mapping. A simple search on [joint HDR and deoising](https://scholar.google.com/scholar?hl=en&q=joint+hdr+denoising) leads to many recent papers on the topics. I suggest the authors consider including such papers in the related work section.
1. Denoising research is at a stage where one has to "pixel peep" to tell the difference. Please provide zoom-in insets of the denoising performance in the result figures.
1. The proposed UHDFour method is only tested on one external dataset, LOL. It's reasonable to include at least another state-of-the-art dataset
1. For the dataset comparison table, the authors should add SID datasets. The argument that SID only works for particular Bayer pattern is not valid. SID data's Bayer or Fuji pattern raw can be demosaiced to RGB raw, meanwhile All the authors' data also undergoes some demosaicing too.

**Summary Of The Paper:**

To tackle the problem of joint luminance enhancement and denoising in low-light imaging, the author proposed 1) a new network architecture, UHDFour, that utilizes authors' observations on noise and signal luminance's relation in the amplitude and phase images in the frequency domain, and 2) a new dataset, UHD-LL, that contains high-resolution 4K low-light / normal light pairs.

**Summary Of The Review:**

Overall, this is a solid paper with interesting insights and new datasets. If the authors and properly address the concerns in the weakness section, I'm happy to upgrade the rating from marginally accept to accept.

---

> ### Author Response · Authors · 2022-11-17
> **Response to Reviewer ohoL**
>
> We sincerely thank you for reviewing our paper and providing us with valuable feedback. We have addressed your concerns below. The revised parts have been marked in blue in the updated paper and Appendix.
>
>
> > **The author should consider adding more ablation studies in the final version.**
>
> Thank you for your suggestion. We have added the suggested ablation studies in Table 5 of the updated paper. As shown, replacing the Fourier branch with the spatial branch with comparable size cannot achieve comparable performance with the full model, showing the efficacy of the Fourier branch. More descriptions and discussions can be found in Section 4.3 of the updated paper.
>
> > **What are the computational and time costs of FFT/IFFT-related operations?**
>
> As the FFT/IFFT operations are used for processing different scales 8$\times$, 16$\times$, and 32$\times$ features in our network, we separately provide their FLOPs and running time in Table 7 of the updated Appendix.
>
> > **HDR scenes are not considered in this paper. I suggest the author mention such drawbacks of the proposed dataset in discussion and/or indication of the dynamic range of the proposed datasets.**
>
> Thank you for your suggestion. As mentioned, we used two cameras, i.e., a Sony α7 III camera and a Sony Alpha a6300 camera to capture our dataset.
>
> Like some existing RGB-based low-light datasets, we save the captured as 8bit sRGB images. We have mentioned this in Section 3 and discussed the limitations in Section 5 of the updated paper.
>
> > **I suggest the authors consider including HDR papers in the related work section.**
>
> As suggested, we have included the HDR related works in Section A of the updated Appendix.
>
> > **Please provide zoom-in insets of the denoising performance in the results figures.**
>
> Besides Figures 6 and 7 in the paper, we have provided the zoom-in insets in Figures 14, 15, 16, 17, 18, 19, 20, and 21 in the updated Appendix.
>
> > **It is reasonable to include at least another state-of-the-art dataset.**
>
> Thank you for your suggestion. As for real low-light image enhancement in the RGB domain, besides our dataset, the LOL is a widely used dataset. We have compared our method with other methods on LOL-v1 dataset. As suggested, we also conducted another comparison on LOL-v2 dataset, which is the dataset collected by the same authors as LOL-v1 and includes more data and scenes.
>
> We have added the quantitative comparison in Table 4 of the updated paper and the visual comparison in Figure 21 of the updated Appendix.  Even though our goal is not to pursue state-of-the-art performance on the LOL-v1 and LOL-v2 datasets, our method achieves satisfactory performance. More details can be found in Section 4.2 of the updated paper and Section G of the updated Appendix.
>
> > **For the dataset comparison table, the authors should add SID dataset.**
>
> As suggested, we have added the SID dataset to Table 1 and explained the differences in Section 3 of the updated paper.
>
> We wish to emphasize that the images in the SID dataset are captured in extremely dark scenes. Its diversity of darkness levels and scenes of SID dataset is limited. Besides, when these RAW data with extremely low intensity is transformed to sRGB image, some information would be truncated due to the bit depth constraints of 8bit sRGB image. In this case, it is challenging to train a network for effectively mapping noise and low-light images to clear and normal-light image using these sRGB images as training data. Note that in the SID paper, to visualize the dark input images, the authors amplify them using a known luminance adjustment ratio before transforming RAW data to sRGB domain by the ISP pipeline.
>
>
> > **Detailed network layer description or table, or source code instead, should be provided.**
>
> We will release our code and the dataset. We will also provide the ablated models and the models pre-trained on different datasets.

---

> ### Author Response · Authors · 2022-12-13
> **Kind Reminder to Reviewer ohoL**
>
> Dear Reviewer,
>
> Thank you again for your constructive feedback. We would like to kindly remind you to confirm whether our changes solved your concerns. If you have any further questions, do not hesitate to reply to our responses. Thank you again for all the efforts that you have made.
>
> If your concerns were solved, could you please consider upgrading the rating? Thank you!
>
> The Authors

---

### Author Response · Authors · 2022-11-18
**Kind Reminder**

We appreciate all the reviewers for their valuable time and constructive comments. The comments help us to substantially improve the quality of our paper.

In our revised paper and the response, all the issues that the reviewers raised have been addressed. The changes we made are marked in blue in the revised paper and Appendix. The revisions and response can be summarized as follows:

- We have added more comparative results, including comparisons with traditional methods and comparisons on extra datasets.
- We have added and tested more ablated models to support our main claims.
- We have provided more details of our method and experiments, including the visual results with zoom-in insets, visual results of amplitude and phase in our network, and the computational and time costs of FFT/IFFT operations.
- We have explained our observations from the FFT aspects and provided more details of our network designs.
- We have discussed the differences between our work and FFT-related low-level vision tasks.
- We have added the introduction of HDR-related works.
- We have answered the minor concerns or suggestions on the details or settings in our paper
- Code and the dataset will be released. We will also provide the ablated models and the models pre-trained on different datasets.

Since it is close to the end of the discussion period, we would like to kindly remind the reviewers to confirm whether our changes solved your concerns. If you have any further questions, do not hesitate to reply to our responses. Thank you again for all the efforts that you have made.

Best wishes,

Authors of paper Embedding Fourier for Ultra-High-Definition Low-Light Image Enhancement

---

### Decision · Program_Chairs · 2023-01-20

**Decision:**

Accept: notable-top-5%

**Justification For Why Not Higher Score:**

N/A

**Justification For Why Not Lower Score:**

In many ways this paper sets a gold standard of what a good paper should be: new & challenging problem with real impact, novel solutions, new dataset, thorough evaluation, extensive comparisons, and good writing. It's going to have a very positive impact to the community.

**Metareview: Summary, Strengths And Weaknesses:**

The paper introduced a new task of UHD low-light image enhancement, and proposed to embed Fourier transform into a cascade network to address the limitations of previous methods. The authors also created a new UHD LLIE dataset for systematic evaluation. Sufficient experiments (compared to many existing low-light enhancement algorithms) and ablation studies demonstrated superior results.

Out of 4 reviewers, 1 (R #ohoL) is positive, and the other 3 are very positive. The very positive reviewers liked the new task, novel solution, new dataset, systematic analysis, outstanding performance and sufficient experiment & analysis (R #Fqm7, #yq2M), and the paper is written well and easy to follow, and the code & dataset will be released (R #aXaB). The AC agrees with these positive points: the problem is new and difficult, yet the solution is novel, effective and efficient.

R #ohoL raised the issue of whether 2xSB could be similar to FB+SB (and hence to discount FB). The authors responded with an updated Table 5 on page 9 showing the advantage of FB+SB. As to the raised issue of HDR with UHD, the AC does not agree with R #ohoL that they always co-occur (many 4K displays are not HDR); besides, we can be less picky for the first attempt to a new problem. The other 3 very positive reviewers raised some minor issues and were addressed in the rebuttal.

In summary, this is a well-written paper that could have real impact on the imaging and computational photography community.


**Note From Pc:**

if the above contains the word "oral" or "spotlight" please see: "oral" presentation means -> notable-top-5% and "spotlight" means -> notable-top-25%. As stated in our emails, we are disassociating presentation type from AC recommendations